# Ultra-wide-field imaging Mueller matrix spectroscopic ellipsometry for semiconductor metrology

Juntaek Oh[1], Jaehyeon Son[1], Changhyeong Yoon[1], Eunsoo Hwang[1], Jinwoo Ahn[1], Jaewon Lee[1], Jinsoo Lee[1], Jiyong Shin[1], Donggun Lee[1], Seunga Lim[1], Jeongho Ahn[2], Younghoon Sohn[3], Sangjin Hyun[4], Myungjun Lee[1] & Taeyong Jo[1] ✉

We propose an ultra-wide-field imaging Mueller matrix spectroscopic ellipsometry (IMMSE) system for semiconductor metrology. The IMMSE system achieves large-area measurements with a 20 mm × 20 mm field of view (FOV)—the largest FOV reported to date—and a spatial resolution of 6.5 μm. It enables the acquisition of over 10 million Mueller matrix (MM) spectra within the FOV, while a unique signal correction algorithm ensures spectrum consistency across the FOV. Leveraging this numerous MM spectra and machine learning, spatially dense metrology across the entire wafer area is achieved. This approach provides over 1987 times more metrology data and 662 times higher throughput compared to conventional point-based methods, such as scanning electron microscopy. We experimentally demonstrate the potential of the IMMSE for yield enhancement in semiconductor manufacturing by identifying spatial variations of dynamic random access memory (DRAM) structures within individual chips as well as across the wafer.

As semiconductor device features have been continuously scaled down, ensuring accurate manufacturing process control has become increasingly challenging[1,2]. The narrow dimensional tolerances lead to local structural variations within a chip, which impacts electrical performance and potentially results in device failure[3,4]. To effectively monitor these variations across an entire wafer, advanced semiconductor metrology techniques with higher spatial density and faster throughput are essential[5–7].

Spectroscopic analysis techniques, such as spectroscopic reflectometry (SR) and spectroscopic ellipsometry (SE), provide essential information on thickness and critical dimensions (CD) due to their high sensitivity and non-destructive nature[8–14]. Mueller matrix spectroscopic ellipsometry (MMSE) further extends these capabilities by utilizing Mueller matrix (MM) to characterize the complex structures of modern semiconductor devices, including overlay errors[15–17].

Despite their advantages, SR, SE, and MMSE share the inherent limitation of point-by-point measurement, which makes it challenging to achieve high spatial density or faster throughput for wafer-scale analysis[18–20]. Scanning electron microscopy (SEM), offers an alternative solution for the characterization of complex device structures with its exceptional resolution[21–23]. However, its small field of view (FOV) limits the identification of structural variations within individual chips as well as comprehensive wafer-scale metrology[24–27].

Imaging-based techniques, such as imaging reflectometry, imaging ellipsometry and imaging MM ellipsometry have been proposed to overcome the limitations of conventional point-based approches[20,28–32]. However, their performance is constrained by a limited FOV of only a few millimeters and the restricted utilization of MM components[33,34]. These limitations restrict rapid metrology across the entire wafer and the precise characterization of complex device

[1]Advanced Process Development Lab 4, Semiconductor R&D Center, Samsung Electronics Co., Ltd., 1-1 Samsungjeonja-ro, hwaseong-si, Gyeonggi-do 18848, Republic of Korea. [2]DRAM Process Development Team, Process Development Department, Semiconductor R&D Center, Samsung Electronics Co., Ltd., 1-1 Samsungjeonja-ro, hwaseong-si, Gyeonggi-do 18848, Republic of Korea. [3]Metrology and Inspection Team, Samsung Electronics Co., Ltd., 1-1 Samsungjeonja-ro, hwaseong-si, Gyeonggi-do 18848, Republic of Korea. [4]Advanced Process Development Team, Semiconductor R&D Center, Samsung Electronics Co., Ltd., 1-1 Samsungjeonja-ro, hwaseong-si, Gyeonggi-do 18848, Republic of Korea. ✉e-mail: ty.jo@samsung.com

structures. Recently, new metrology techniques have been explored to address current technical limitations, including line scan methods employing hyperspectral imaging[35], pupil ellipsometry with self-interferometry[36], and nano-spot spectroscopy using microspheres[37,38].

In this study, we propose an ultra-wide-field imaging Mueller matrix spectroscopic ellipsometry (IMMSE) system to overcome the limitations of current metrology techniques for semiconductor manufacturing[39,40]. The IMMSE system is capable of measuring MM spectra from all pixels of a hyperspectral image cube, covering a 20 mm × 20 mm area with 3200 × 3200 pixels and a pixel resolution of 6.5 μm. This implies over 10 million MM spectra can be obtained in the hyperspectral image cube. To ensure spectrum consistency over the wide FOV, we propose a method for calculating the MM spectrum that incorporates in-situ correction of polarization-dependent transmittance variations in the entire optical system.

By leveraging the numerous MM spectra and machine learning algorithm, we achieved high spatial density metrology across the entire wafer. Compared to the conventional point-based techniques, the IMMSE provided over 1987 times more metrology data and 662 times faster throughput per measurement point.

Utilizing its capability for the spatially dense metrology, we experimentally demonstrated that the IMMSE system could serve as a potential solution for the rapid identification of the defective chips during the semiconductor manufacturing. We presented the measurement results of thin film thickness, CD, and overlay errors in dynamic random access memory (DRAM) device. These results offered significant insights for optimizing semiconductor manufacturing processes by identifying spatial variations in DRAM device structures within individual chips as well as across the wafer. Furthermore, we verified its capability for statistical analysis of overlay errors within chips to detect defective chips based on the numerous metrology data points.

## Result

### Ultra-wide-field imaging Mueller matrix spectroscopic ellipsometry (IMMSE) system

For the experimental demonstration of the proposed ultra-wide-field IMMSE, we utilized a custom-designed optical system. Figure 1a shows the schematic of the proposed IMMSE system, comprising three main modules: illumination module, imaging module, and polarization manipulation module.

A plasma-based broadband light source was employed to cover the system's wide wavelength range, extending from ultraviolet to infrared. A custom-designed monochromator allowed the selection of monochromatic light and it was delivered to the illumination optics using an optical fiber. The illumination optics was designed for the uniform illumination over the wide measurement area with the angle of incidence (AOI) of 65°. An engineered diffusing glass and telecentric lens relay were configured to achieve Köhler illumination and uniform illumination properties throughout the wide illumination area, respectively.

The imaging module consisted of imaging optics and image sensor. The imaging optics was implemented using two concentric spherical mirrors, commonly known as the Offner configuration, to minimize chromatic aberration across the measurement area and wavelength range[41–43]. The signal was captured using an image sensor and the sensor's position was precisely adjusted to focus the tilted image, following the Scheimpflug principle.

The polarization manipulation module was consisted of a polarization state generator (PSG) and a polarization state analyzer (PSA). The PSG controls the polarization state of the illumination light, while the PSA analyzes the sample's polarization sensitivity by changing the polarization of the reflected light from the sample. The PSG and PSA modify the polarization of the light by rotating the wire-grid polarizers.

The alignment optics, based on a conventional microscope, was used to ensure precise measurement positioning and image focusing. Figure 1b illustrates the realized optical setup. For the experimental demonstration, we developed a metrology equipment that was integrated with the optical setup, a wafer transport robot, and a stage system for wafer movement (See Methods for detailed information about the optical setup).

Figure 1c shows the data acquisition and processing flow of the proposed IMMSE system. In this study, we acquired a hyperspectral image cube with an ultra-wide field of view (FOV) of 20 mm × 20 mm under multiple wavelength conditions. Given its high spatial resolution of 3200 × 3200 pixels and a pixel size of 6.5 μm, the hyperspectral cube contains over 10 million individual intensity spectra in a single acquisition. Utilizing the polarization manipulation module, hyperspectral image cubes were measured under multiple polarization conditions. The obtained signals were calibrated using unique data processing algorithm to correct for signal distortion induced by the measurement system, including rotating polarizer and aberrations, such as image distortion. The MM spectrum can be calculated for all pixels in the calibrated image by analyzing the signal variations corresponding to the polarization changes. Using the spatially resolved Mueller matrix (MM) spectra, a regression model was trained with reference data from conventional metrology tools such as SEM. This approach enabled high-spatial-density, wafer-scale metrology of nanostructures.

### System-induced signal distortion

One of the key features of the proposed wide-field IMMSE system for semiconductor manufacturing is its ability to deliver consistent and reliable measurement results over the FOV. Specifically, measurements taken at the same point on a sample should remain consistent, whether the point is located at the center or the edge of the FOV. However, in practice, signal distortion occurs depending on the positions within the FOV.

The measurement discrepancies within the large FOV are expected to arise from the non-uniformity of optical performance over their operational regions, such as the transmittance or reflectance of lenses, mirrors, and polarizers. Specifically, the signal recorded at each pixel is the result of light propagated through different regions of optical components. These components exhibit position- and wavelength-dependent variations in transmittance and reflectance, which leads to pixel-wise discrepancies in the measured intensity. In addition to the performance inconsistency of the optical components, the signal also changes with the rotation of the polarizer, as the light passes through different areas of polarizer by the rotation, as shown in Fig. 2a.

Previous research on the imaging ellipsometry addressed the inhomogeneity of retardation in rotating compensator by employing an extra phase constant in the equation[33,34]. In this work, we introduce in-situ correction method of polarization-dependent transmittance variations to obtain reliable signals across the FOV by eliminating system-induced signal distortion in the dual rotating polarizer configuration. This was achieved by employing condition-specific relative transmittance (RT) constant and compensating for the performance changes in the PSG, PSA, and wavelength. In addition to polarization-induced intensity variations, residual geometric distortion from the imaging optics can degrade the spatial alignment of hyperspectral data across the wide FOV. Although the Offner-type reflective relay used in the imaging module minimizes chromatic and geometric aberrations, slight spatial distortion remains due to practical implementation factors. To address this, we applied a geometric distortion correction during post-processing. A reference target was used to measure the pixel-wise displacement vectors across the field, and these vectors were used to remap each image to a common spatial grid. In semiconductor metrology, even a few micrometers of spatial misalignment can affect spectral accuracy, especially when analyzing pixel-wise variations across polarization conditions. Our correction ensures

spatial consistency across wavelengths and polarization states, supporting accurate MM reconstruction at high spatial resolution.

Furthermore, while the camera manufacturer provides basic linearity correction, some residual nonlinear response remains, particularly in low-intensity regions. This is critical in our system due to large intensity variation across polarization states. Additional calibration was performed using a bare silicon wafer under 500 nm illumination, with the exposure time incrementally increased by 0.015 ms. A separate dark measurement was also taken. The resulting intensity curve was corrected using spline interpolation to generate a refined response map.

### System factor analysis

The intensity and partial polarization characteristics of light are described using the Stokes vector. The relationship between the light before and after interacting with the sample can be expressed as follows:

$$\mathbf{s}_{out} = \mathbf{M}_1 \mathbf{M}_2 \cdots \mathbf{M}_n \mathbf{s}_{in} \tag{1}$$

where $\mathbf{s}_{in}$ and $\mathbf{s}_{out}$ are Stokes vectors of input and output light, respectively, $\mathbf{M}_1, \mathbf{M}_2, \cdots, \mathbf{M}_n$ are the MMs of optical components in the system. Since the dual rotating polarizer configuration was employed in this study, the MM is represented as a $3 \times 3$ square matrix, excluding circular polarization characteristics[44]. Based on Eq. (1), the intensity measured using the proposed system can be expressed as a matrix product of MMs of the polarization manipulation components and the sample wafer, as shown below:

$$L_{ij} = \mathbf{d}^{\mathsf{T}} \mathbf{Q}_j \mathbf{M} \mathbf{P}_i \mathbf{s} \tag{2}$$

where $L_{ij}$ is the measured intensity at the $i$-th PSG condition and the $j$-th PSA condition. $\mathbf{M}$ represents the MM of the sample wafer, $\mathbf{P}_i$ and $\mathbf{Q}_j$

denote the MMs of the PSG and the PSA, respectively. $\mathbf{s}$ is Stokes vector of illumination light and $\mathbf{d}$ represents the cumulative polarization sensitivity of all detection-side components after the PSA. This includes the Offner relay, sensor window, and the sCMOS image sensor, all modeled as MMs. To incorporate the RT of the optical system, we reformulated both polarizer MMs $\mathbf{P}_i$ and $\mathbf{Q}_j$ as follows:

$$\mathbf{P}_i = \zeta_i \begin{bmatrix} 1 & \cos 2\theta_{\alpha,i} & \sin 2\theta_{\alpha,i} \\ \cos 2\theta_{\alpha,i} & \cos^2 2\theta_{\alpha,i} & \cos 2\theta_{\alpha,i} \sin 2\theta_{\alpha,i} \\ \sin 2\theta_{\beta,j} & \cos 2\theta_{\alpha,i} \sin 2\theta_{\alpha,i} & \sin^2 2\theta_{\alpha,i} \end{bmatrix} = \zeta_i \mathbf{p}_i \cdot \mathbf{p}_i^{\mathsf{T}} \tag{3}$$

$$\mathbf{Q}_j = \xi_j \begin{bmatrix} 1 & \cos 2\theta_{\beta,j} & \sin 2\theta_{\beta,j} \\ \cos 2\theta_{\beta,j} & \cos^2 2\theta_{\beta,j} & \cos 2\theta_{\beta,j} \sin 2\theta_{\beta,j} \\ \sin 2\theta_{\beta,j} & \cos 2\theta_{\beta,j} \sin 2\theta_{\beta,j} & \sin^2 2\theta_{\beta,j} \end{bmatrix} = \xi_j \mathbf{q}_j \cdot \mathbf{q}_j^{\mathsf{T}} \tag{4}$$

where $\mathbf{p}_i = \begin{bmatrix} 1 & \cos 2\theta_{\alpha,i} & \sin 2\theta_{\alpha,i} \end{bmatrix}^{\mathsf{T}}$ and $\mathbf{q}_j = \begin{bmatrix} 1 & \cos 2\theta_{\beta,j} & \sin 2\theta_{\beta,j} \end{bmatrix}^{\mathsf{T}}$ are the polarizer vectors, $\theta_{\alpha,i}$ and $\theta_{\beta,j}$ are the $i$-th rotation angle of the PSG and the $j$-th rotation angle of the PSA, respectively. $\zeta_i$ and $\xi_j$ indicate the RT of the rotating PSG and PSA, respectively. Note that both rotation angles, $\theta_{\alpha,i}$ and $\theta_{\beta,j}$, are evenly spaced over 360°. Using the Eqs. (3) and (4), Eq. (2) can be rearranged as shown in Fig. 2b, transforming the intensity equation into a product of scalars, as follows:

$$L_{ij} = \eta_o \alpha_j K_{ij} \beta_i \tag{5}$$

where $\eta_o$ denotes the intensity constant, $\alpha_j = \mathbf{d}^{\mathsf{T}} \mathbf{q}_j \xi_j$ indicates PSA-side total RT, defined as the PSA factor, $\beta_i = \zeta_i \mathbf{p}_i^{\mathsf{T}} \mathbf{s}$ represents PSG-side total RT, defined as the PSG factor, and $K_{ij} = \mathbf{q}_j^{\mathsf{T}} \mathbf{M} \mathbf{p}_i$ denotes the kernel factor. The system factors, PSA factor ($\alpha_j$) and PSG factor ($\beta_i$), can be

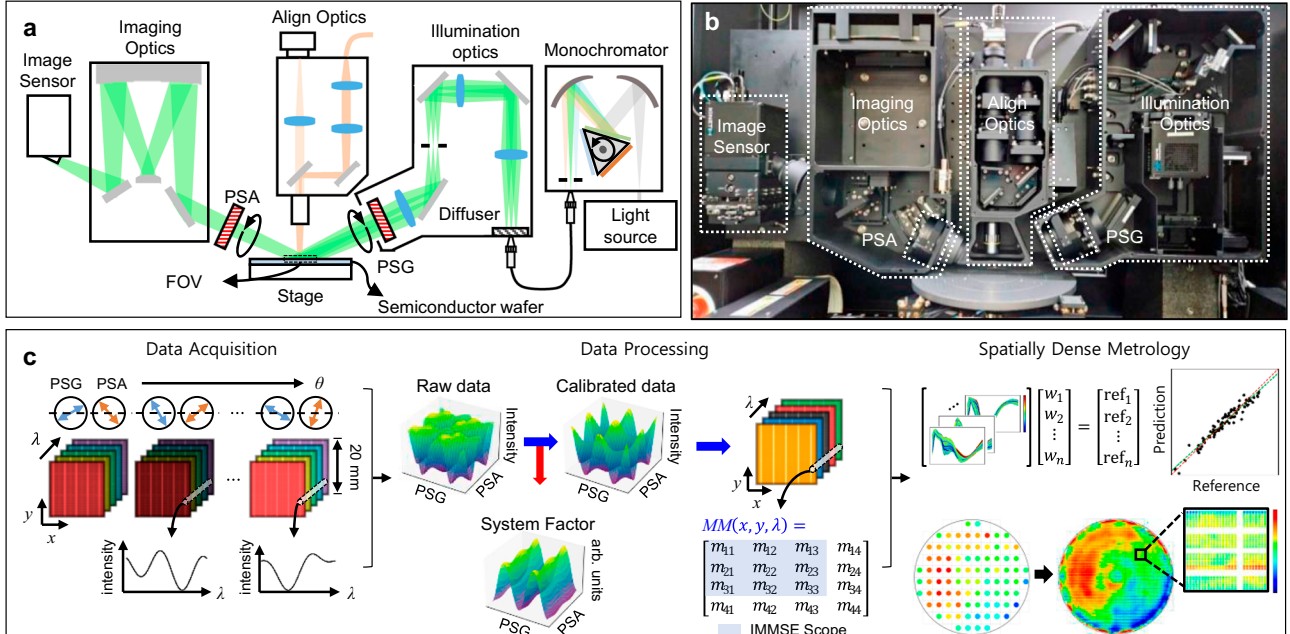

**Fig. 1 | Configuration of the IMMSE system and its process flow. a** Schematic of the IMMSE system, which consists of light source, monochromator, illumination optics, imaging optics, image sensor, polarization manipulation module (indicated as PSG and PSA) and align optics. **b** Photograph of the realized optical setup implemented within the metrology tool, which also incorporates a wafer transport robot and a stage system for wafer handling. **c** Data acquisition and processing flow of the IMMSE system. Hyperspectral image cubes under multiple polarization conditions were acquired. The system-induced signal distortion, referred to as the system factor, was corrected in the acquired signal. The $3 \times 3$ Mueller matrix (MM) spectrum was derived by analyzing signal variations under different polarization states. The spatially dense metrology was achieved by utilizing a regression model trained on reference metrology data from conventional SEM and the acquired spatially resolved MM spectra.

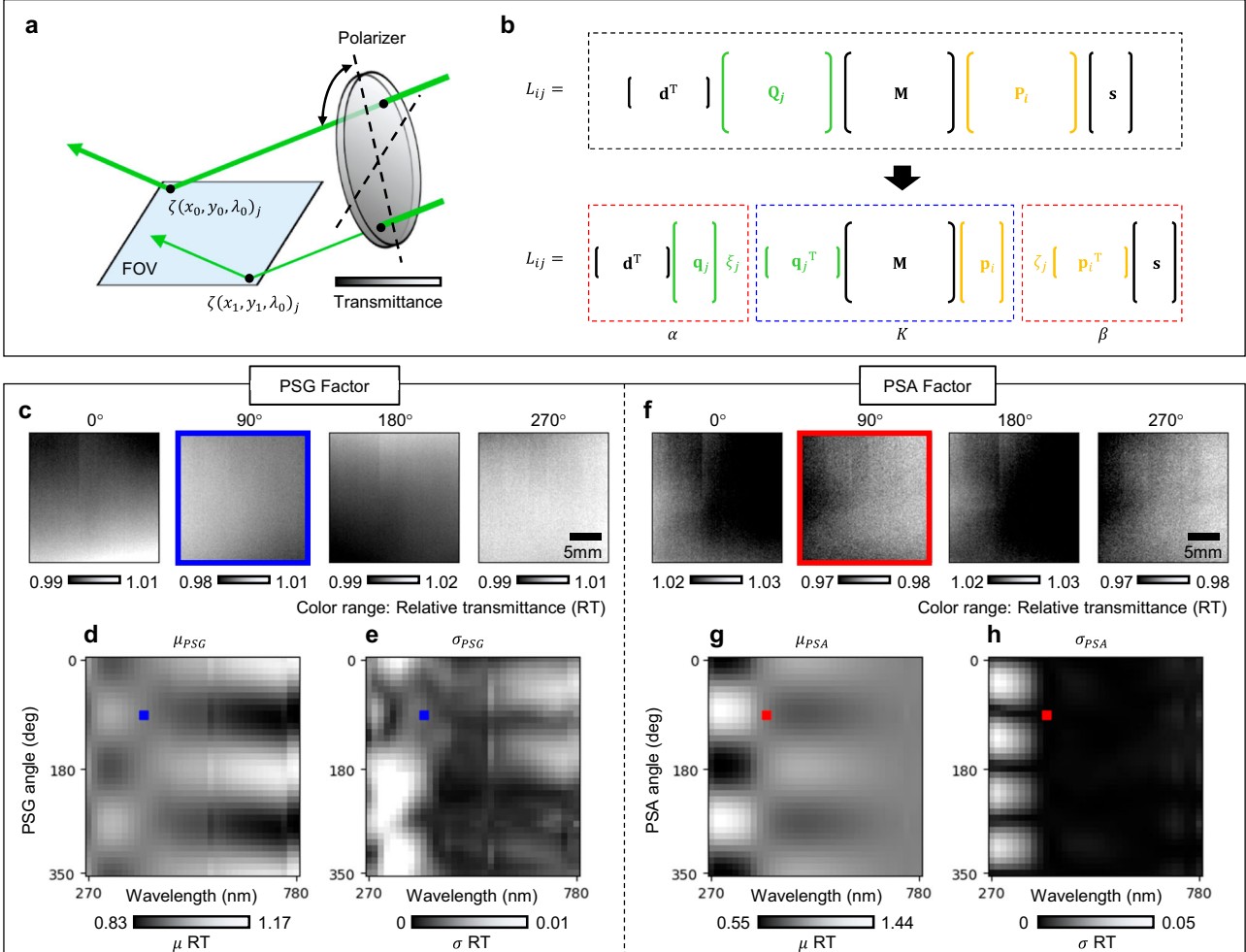

**Fig. 2 | System-induced signal distortion and system factors. a** The schematic of the relative transmittance (RT). Light arrived at each pixel traversed distinct regions of a polarizer, where optical properties such as transmittance vary spatially. Furthermore, these properties are also changed with the rotation of the polarizer. **b** The schematic of matrix reformulation. By utilizing the characteristics that the polarizer MM can be decomposed into two vectors, the measured intensity signal was reformulated as a product of scalars: the PSG, PSA factor, and kernel factor. **c** The obtained PSG factor at a wavelength of 405 nm for rotation angles of 0°, 90°, 180°, and 270°. **d, e** The mean ($\mu$) and standard deviations ($\sigma$) values for PSG factor across all wavelength and polarization conditions. **f** The PSA factor under the same wavelength and polarization conditions as shown in (c). **g, h** $\mu$ and $\sigma$ for PSA factor across all wavelength and polarization conditions.

regarded as random variables that always take positive values. Without loss of generality, the average values of both system factors and the kernel factor are assumed to be 1. The first essential step in determining both system factors is to employ a new variable that the system factors are eliminated, as follows:

$$R_{ij} = \frac{L_{ij}}{\left(\sum_n L_{in}\right)\left(\sum_n L_{nj}\right)} = \frac{K_{ij}}{\left(\sum_n \alpha_n K_{in}\right)\left(\sum_n \beta_n K_{nj}\right)} = \frac{K_{ij}}{H_{\alpha,i} H_{\beta,j}} \quad (6)$$

where $H_{\alpha,i} = \sum_n \alpha_n K_{in}$ and $H_{\beta,j} = \sum_n \beta_n K_{nj}$. By the definition, the kernel factor ($K_{ij}$) is a linear combination of harmonic bases weighted by each MM component. Therefore, $H_{\alpha,i}$ and $H_{\beta,j}$ have simple harmonic form and the kernel factor can be derived from $R_{ij}$ by solving for the harmonic parameters of $H_{\alpha,i}$ and $H_{\beta,j}$. Once the kernel factor is obtained, both system factors, $\alpha_j$ and $\beta_i$, can be determined using Eq. (5). (See Methods for the detailed calculation of the system factors). For the calculation of Eq. (6), hyperspectral images of a bare silicon wafer were measured under full-revolution conditions for both the PSG and PSA, with rotations ranging from 0° to 350° in 10° increments.

Figure 2c shows the calculated PSG factor at a wavelength of 405 nm under four of the 36 rotation conditions, specifically when the PSG was rotated to 0°, 90°, 180°, and 270°. The results indicate the RT variations over the FOV of ~2%, 3%, 2%, and 2%, respectively. To investigate the overall trend across all wavelengths and polarization conditions, the mean $\mu_{psg}$ and standard deviation $\sigma_{psg}$ values of the PSG factor for all optical conditions were calculated, as shown in Fig. 2d, e, respectively. The blue-marked points in Fig. 2d, e represent the result corresponding to the PSG factor at 405 nm and under a 90° rotation condition, as shown in Fig. 2c. The results suggest that the RT averaged over the FOV varies between 83% and 117% for all optical conditions, while the standard deviation of the RT within the FOV varies up to 1%. Figure 2f presents the PSA factor results under the same conditions as Fig. 2c, showing a maximum transmittance difference of up to 1% within the FOV. Figure 2g, h show the results of the mean and standard deviation of the PSA factor across the FOV for all wavelengths and polarization conditions. For the transmittance on the PSA side, unlike that of the PSG, the average RT within the FOV spans a relatively wider range, from 55% to 144%.

## Mueller matrix of bare silicon wafer
For the experimental demonstration of the system-induced signal distortion correction, we measured bare silicon sample wafer. The hyperspectral image cubes were obtained under nine combinations of

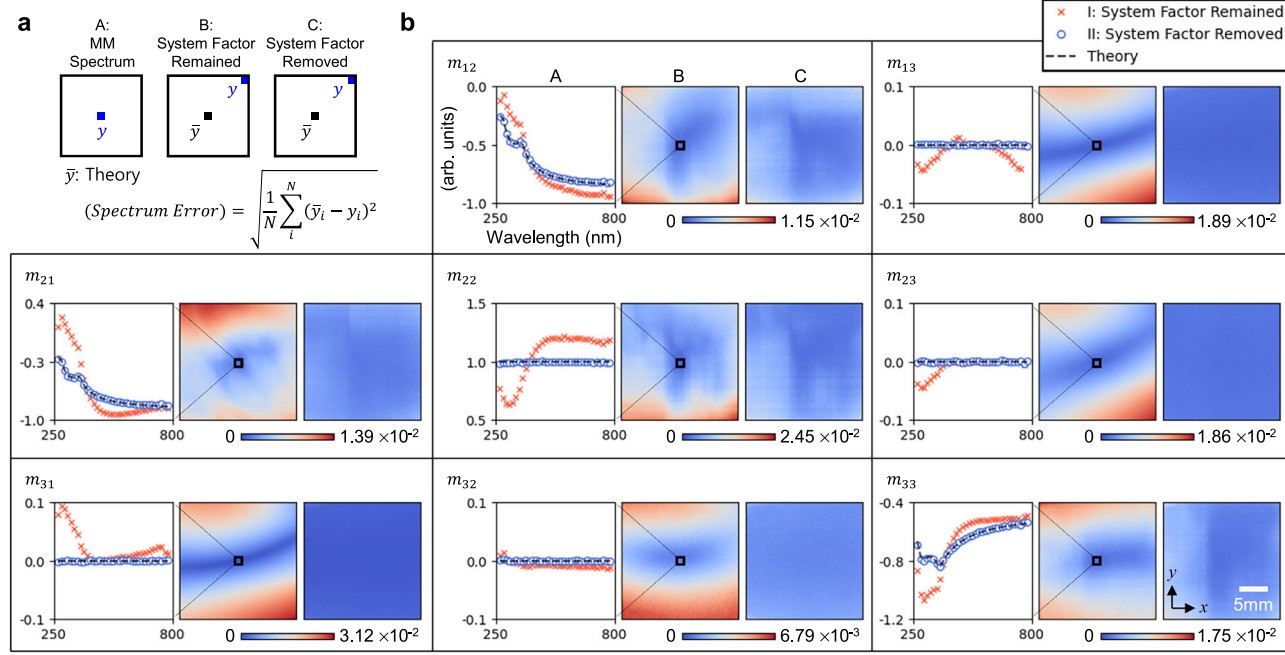

**Fig. 3 | MM of the bare silicon wafer. a** Schematic of data analysis: (A) shows the comparison of the MM spectrum with theoretical values for both (I) before and (II) after the system factors removal. (B) and (C) represent the spectrum errors between the MM spectrum at the FOV center and those in other regions of the FOV. (B) shows the results with system factors remaining, while (C) represents the result after the system factors have been removed. **b** The results of (A), (B), and (C) for all MM elements.

**Table 1 | Spectrum errors with respect to the theoretical spectrum and within FOV**

|   |   | m12 | m13 | m21 | m22 | m23 | m31 | m32 | m33 |
|---|---|------|------|------|------|------|------|------|------|
| A | I | 0.1126 | 0.0204 | 0.2283 | 0.2089 | 0.0160 | 0.0348 | 0.0088 | 0.1214 |
|   | II | 0.0085 | 0.0009 | 0.0086 | 0.0064 | 0.0013 | 0.0009 | 0.0008 | 0.0076 |
| B |   | 0.0018 | 0.0040 | 0.0025 | 0.0039 | 0.0036 | 0.0073 | 0.0014 | 0.0029 |
| C |   | 0.0010 | 0.0003 | 0.0006 | 0.0020 | 0.0002 | 0.0003 | 0.0002 | 0.0015 |

(A) in Fig. 3b and the table shows the spectrum error relative to the theoretical spectrum. (I) and (II) indicate the spectrum error values for both before and after the system factors removal, respectively. The spectrum error variations within the FOV (B) before and (C) after the system factors removal, respectively.

polarization states where the PSG and PSA are rotated with 5:3 rotation angle ratio. The measured intensity dataset was divided by the obtained PSG factor and PSA factors to compensate for the system-induced signal distortions, resulting in the kernel factor. Since the kernel factor is defined as a linear combination of orthogonal bases associated with the MM elements, each MM component can be directly calculated using the following form:

$$m_{xy} = \frac{a_{xy}}{N_\theta} \sum_{t=1}^{N_\theta} K_{5t,3t} B_{xy}(\theta_{5t}, \theta_{3t}) \qquad (7)$$

where $m_{xy}$ represents the MM element in the $x$-th row and the $y$-th column, $a_{xy}$ is the normalizing constant, $B_{xy}$ is the basis vector, $\theta$ is the rotation angle and $N_\theta$ is the number of polarization condition. (See Methods for the detailed calculation of the MM.)

We analyzed the calculated MM result of the bare silicon wafer in three ways as shown in Fig. 3a. (A): A comparison between the MM spectrum obtained at the FOV center and the theoretical spectrum calculated using the Fresnel equation[45]. (B), (C): Spectrum error maps relative to the FOV center for both before and after the system factors removal, respectively. In this study, root-mean-squared-error (RMSE) was utilized to evaluate the spectrum error. Since bare silicon is an isotropic material, the off-diagonal components of the MM should

ideally be zero. Any nonzero values indicate the residual of the system-induced signal distortions. For the off-diagonal components, such as $m_{13}, m_{31}, m_{23}$, and $m_{32}$, the MM values exhibited approximately zero after the system factors were removed. In addition, the diagonal components, such as $m_{12}, m_{21}, m_{22}$, and $m_{33}$ agreed well with the theoretical spectrum as shown in (A) of Fig. 3b. (A) in Table 1 presents the summary of spectrum errors relative to the theoretical spectrum, both before and after the system factor removal. The result verified that the spectrum error values were reduced to less than $10^{-2}$ for all MM elements after the system factors were removed.

To demonstrate the spectrum consistency across the FOV, spectrum error maps relative to the spectrum at the FOV center were calculated. (B) and (C) in Fig. 3b represent the spectrum error distributions over the FOV before and after the system factor elimination, respectively. For the result (B), significant spectrum discrepancies were observed between the center and the edge of the FOV. In contrast, for the result (C), the spectrum errors within the FOV were noticeably reduced. For the quantitative evaluation, the standard deviation of the spectrum error maps of all MM components for both (B) and (C) were calculated as shown in Table 1. The results indicate that the dispersions of the spectrum error within the FOV were reduced after the system factors removal, with significant improvements observed for the off-diagonal components.

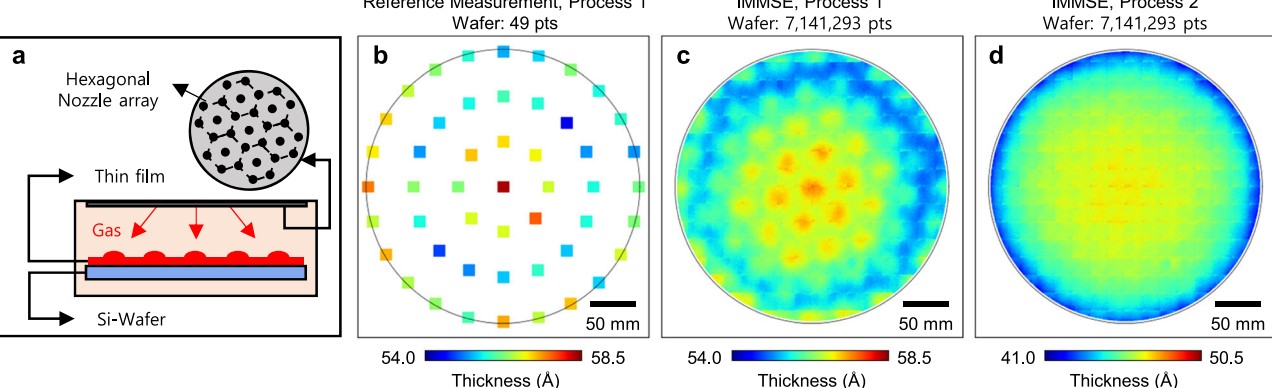

**Fig. 4 | Thin layer thickness metrology of a DRAM wafer and comparative analysis. a** Schematic diagram of thin-film deposition using a gas chamber equipped with multiple nozzles arranged in a hexagonal array. **b** Thickness measurement results of a wafer processed under the initial CVD condition (designated as Process 1), obtained using a conventional point-based ellipsometry technique. This method reveals only limited information about the overall thickness distribution. **c** IMMSE-based thickness measurement of the same wafer, showing distinct non-uniformity patterns across the full wafer area, which are presumed to originate from the characteristics of the nozzles array. **d** IMMSE measurement results for the wafer processed under revised process conditions (Process 2), showing improved film thickness uniformity across the wafer.

## DRAM metrology and applications

We demonstrated that the proposed IMMSE technique provides reliable and consistent MM spectra over an ultra-wide FOV. To enable spatially dense metrology based on these spectra, we employed a machine learning algorithm—specifically, Ridge regression. The regression model was trained using both the acquired massive number of MM spectra and reference data obtained from conventional point-based metrology tools such as SEM. This approach allowed us to achieve wafer-scale characterization of nanostructures, with significantly higher spatial sampling density compared to traditional methods (See "Data acquisition procedure" and "Machine learning for spatially dense metrology" for the detailed information).

In this section, we present practical applications of the IMMSE in DRAM manufacturing, demonstrating its effectiveness in analyzing structural variations and detecting process-induced deviations across the wafer with high spatial resolution.

Chemical vapor deposition (CVD) is one of the essential manufacturing processes in semiconductor fabrication, used to form thin layers on the surface of semiconductor wafers[46,47]. As illustrated in Fig. 4a, the CVD schematic shows thin-film deposition within a gas chamber with multiple nozzles arranged in a hexagonal array, which can affect the spatial thickness uniformity during deposition.

Figure 4b presents the thickness measurement results of a wafer processed under a specific CVD condition (indicated as Process 1). The measurements were acquired at 49 points across the wafer using a conventional point-based ellipsometry system, which reveals only coarse global thickness trends due to its sparse spatial sampling.

In contrast, the thickness measurement of the same sample wafer using the IMMSE reveals distinct, hexagonal patterns throughout the wafer, as shown in Fig. 4c. The spatially dense metrology, with >7 million points, enables the identification of unique thickness variations that the conventional point-based method fails to detect. These patterns could be interpreted as fingerprints of the manufacturing equipment and offer invaluable insights for the process optimization to enhance the yield.

Figure 4d illustrates the measurement results of a wafer processed under modified CVD conditions (referred to as Process 2). Unlike Fig. 4c, d exhibits no distinctive patterns, implying that the modified process conditions are better suited for this manufacturing step.

The DRAM capacitor structure features a high aspect ratio trench filled with metallic materials (Fig. 5a), which is critical for achieving the desired electrical features within a limited cell footprint. Due to this geometry, precise control of the etch process before the metal fill is essential to ensure uniform depth and profile across the wafer. In particular, the CD measured at the top surface after etching serves as an indirect but effective indicator for assessing the quality and uniformity of the deep etch process[48] (Fig. 5b).

Figure 5 presents a comparative analysis of CD measurements obtained using the conventional SEM and the proposed IMMSE system for DRAM wafers processed under two different etch conditions. For the wafer processed under the regular etch process referred to as Process 1, the SEM-based CD measurements (Fig. 5c) are limited to 536 points, which is insufficient to capture global CD variation trends. In contrast, the IMMSE-based measurements (Fig. 5d) provide over 10 million CD values across the full wafer, revealing concentric CD variation patterns (black dashed circles in Fig. 5d) that remain hidden in the SEM results due to sparse sampling. It is noteworthy that the proposed method can identify CD variation patterns at the sub-nanometer scale.

A revised etch condition (Process 2) was applied to suppress these spatial variations. As seen in Fig. 5e, the SEM result does not adequately reflect changes in global CD distribution. However, the IMMSE result (Fig. 5f) confirms that the circular variation pattern has been effectively eliminated, and the overall CD uniformity is significantly improved across the wafer. These results demonstrate the capability of the IMMSE system to enable dense, wafer-scale metrology, offering critical insights into etch process performance that are not accessible through conventional SEM-based sampling.

Figure 6a illustrates the DRAM transistor structure and the principle of overlay evaluation. The overlay error measured in this study refers to the misalignment between the bit line (BL) and the contact material (CM) to the drain of the transistor, which are vertically stacked structures. The BL delivers read/write signals, while the CM connects the drain region of the cell transistor to the BL. Misalignment between these two layers can lead to incomplete or unstable electrical contact, potentially resulting in functional failure or increased resistance. Therefore, precise overlay control is critical to ensure reliable device operation and high manufacturing yield[48,49].

Figure 6b shows a top-view SEM image corresponding to the black arrow and shaded plane in Fig. 6a. The overlay between the BL and CM is estimated by measuring the relative distance between the two features. Figure 6c presents a side-view TEM image that reveals extremely narrow gaps between adjacent structures. The combination of ultra-small feature size and sub-nanometer misalignment, as illustrated in Fig. 6a–c, poses a significant challenge for overlay metrology, especially in achieving both high sensitivity and wafer-scale coverage.

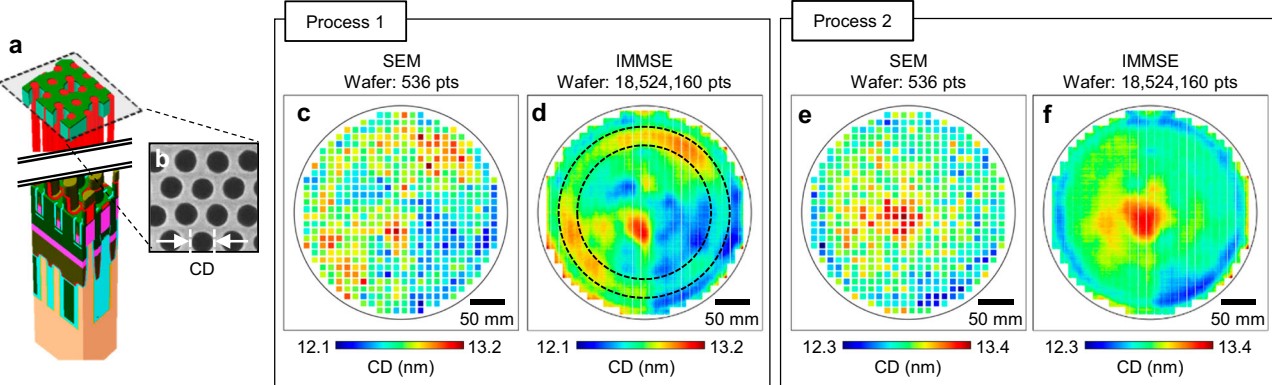

**Fig. 5 | CD metrology of a DRAM wafer and analysis. a** Schematic illustration of the DRAM structure. **b** SEM image corresponding to the cross-sectional plane illustrated as black dashed box in(a). **c** CD measurement result obtained using a conventional SEM. The wafer was processed under the regular etch process referred to as Process 1 in the figure. **d** CD measurement result acquired using the IMMSE. While the SEM provides only 536 measurement points, the IMMSE yields over 10 million CD data points across the entire wafer. This dense spatial sampling reveals distinct circular CD variation patterns (black dashed circles) that are not captured in the SEM results. **e** CD measurement result of the wafer processed under the revised etch process referred to as Process 2 using a conventional SEM. **f** CD measurement result of the same wafer acquired using the IMMSE. The circular CD variation pattern observed in Process 1 has been eliminated, and the overall CD uniformity across the wafer is improved under Process 2.

Figure 6d presents the overlay measurement results obtained using a conventional SEM for a wafer processed under specific lithography conditions (referred to as Process 1), with ~6000 points measured across the wafer. Figure 6e shows an enlarged view of the same SEM data, focusing on four locally adjacent shot regions comprising 261 measurement points.

Figures 6f, g show the IMMSE overlay measurements of the same wafer. While the SEM results provide only 6000 points across the wafer and 261 points within the four localized shot areas, the IMMSE yields >10 million measurements for the entire wafer and over 100,000 measurements within the same localized region.

Leveraging its massive number of overlay measurements, distinct variation patterns on the left side of each shot area can be clearly identified, as indicated by the black arrows in Fig. 6f. These localized variations remain undetected in the SEM results but are readily observed through the dense spatial sampling enabled by IMMSE.

To evaluate the effect of process optimization capability of the IMMSE, a second wafer processed under revised lithography conditions (referred to as Process 2 in the Fig. 6) was analyzed. The IMMSE results for this wafer, shown in Fig. 6h, i, indicate that the shot-level overlay variation pattern is no longer present, suggesting improved lithography performance and greater process maturity compared to Process 1.

In addition to these qualitative analyses of the manufacturing process based on the spatially dense overlay maps, the IMMSE also enables quantitative evaluation of overlay variation within individual chips by utilizing a large number of measurement points. Figure 6j presents a histogram of the overlay distribution within a single chip area outlined in red box in Fig. 6f, i. The result under Process 1 exhibits a mean offset of −1.31 nm with a standard deviation of 0.36 nm, whereas Process 2 yields a near-zero mean value of −0.03 nm and a reduced standard deviation of 0.24 nm.

The obtained statistical analysis can be applied to any localized region on the wafer. Figure 6k illustrates the distribution of the mean and standard deviation of overlay values computed from all cell blocks within each of 100 chips located in the regions shown in Fig. 6f and i. The result indicates that the distribution of both metrics, mean and standard deviation, in Process 2 are significantly improved compared to those in Process 1.

## Discussion
In this study, we demonstrated an ultra-wide-field imaging Mueller matrix spectroscopic ellipsometry (IMMSE) system for semiconductor metrology, capable of acquiring over 10 million spatially resolved $3 \times 3$ Mueller matrix (MM) spectra across a $20\,mm \times 20\,mm$ field of view (FOV) with 6.5 μm spatial resolution.

For the wide-FOV metrology, ensuring spectral consistency across the entire FOV area is critical. We achieved high spectrum consistency across the ultra-wide FOV, suppressing the spectrum error relative to the theoretical MM spectrum of bare silicon wafer by 94% and spectral variation across the FOV by 73%, compared to the conventional MM calculation method. To enable this, we utilized a unique correction algorithm based on relative transmittance—referred to as system factors—which compensates for spatial non-uniformities of the signal induced from optical aberration and rotating polarizers.

Based on the high spectrum consistency across the FOV and the incorporation of a machine learning algorithm, the IMMSE enabled spatially dense metrology, revealing over 10 million metrology data across the wafer. Building upon these capabilities, practical applications of the IMMSE in DRAM manufacturing were suggested.

The experimental demonstrations were conducted on 12-inch DRAM wafers sourced from a real-world semiconductor manufacturing fab, where the critical features are notably small and process variations are difficult to detect. The results highlight the IMMSE system's practical applicability and high sensitivity.

The thin film thickness, critical dimensions (CD), and overlay misalignment errors of the DRAM wafer were measured using the propose method. The IMMSE enabled the detection of distinctive spatial variation patterns across the wafer, which can be effectively suppressed under revised process conditions. These results demonstrate the effectiveness of the IMMSE in accurately monitoring and controlling process-induced pattern variations.

Additionally, statistical analysis of the overlay data enabled the identification of potentially defective chips exceeding the overlay thresholds. By employing the proposed technique, process abnormalities can be detected quickly and well before the electrical test, enabling rapid process feedback and contributing to yield improvement. These results confirm that the IMMSE system can serve as a powerful platform for evaluating and optimizing semiconductor manufacturing processes, ultimately contributing to yield enhancement.

The regression model employed in this study demonstrated high prediction accuracy, with root mean square error (RMSE) values below 1 nm for all evaluated parameters—including thin film thickness, CD, and overlay—as summarized in Table 2. This sub-nanometer accuracy indicates that the IMMSE system is capable of resolving nanometer-

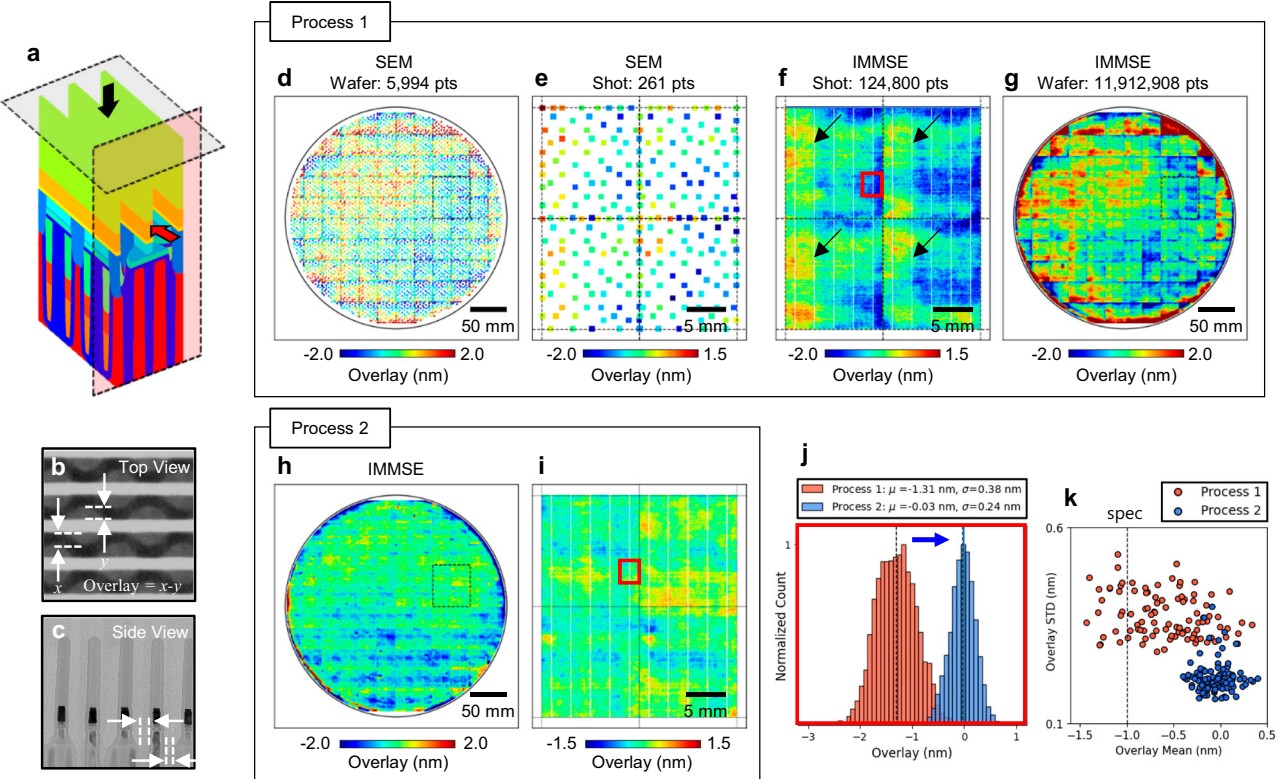

**Fig. 6 | Overlay metrology of a DRAM transistor structure and comparative analysis between conventional SEM and IMMSE. a** 3D schematic of the DRAM transistor structure. **b** SEM image showing the BL; horizontal lines and CM; circular patterns beneath the BL from a top-down view, corresponding to the black arrow and shaded plane in (**a**). **c** TEM image of the same structure viewed from the side, as indicated by the red arrow and shaded plane in (**a**). The overlay error is evaluated by measuring the vertical misalignment between the BL and CM layers. **d** Overlay measurements acquired using a conventional SEM across the wafer processed under Process 1. **e** Enlarged view of four adjacent shot regions indicated by the dashed box in (**d**), showing the local overlay distribution in more detail. **f** IMMSE overlay measurement results of the same region shown in (**e**), revealing recurring overlay variation patterns within each shot, as marked by black arrows. **g** Full-wafer IMMSE measurement results of the same wafer shown in (**d**), providing high-density overlay information for Process 1. **h, i** IMMSE overlay measurements of a different wafer processed under modified conditions (Process 2), demonstrating improved overlay performance and reduced variation. **j** Histogram of overlay values within a single chip area, marked by red boxes in (f) and (i), comparing Process 1 and Process 2 results. **k** Statistical distribution of the mean and standard deviation of overlay values for 100 chips within the regions marked in (f) and (i), demonstrating improved overlay control under Process 2.

scale structural variations, which is essential for monitoring and controlling process-induced fluctuations in advanced semiconductor fabrication.

While the required accuracy may vary across applications, the achieved sub-nanometer precision is consistent with the metrology demands of next-generation semiconductor technologies, which continue to scale toward smaller nodes with tighter process tolerances[50].

As a further evaluation of system performance, we assessed repeatability of the system by conducting repeated overlay measurements at the same wafer location. The resulting 3-sigma repeatability, averaged across all measurement points within a single shot, was -0.36 nm. This indicates stable and consistent overlay measurement capability, making the IMMSE system suitable for high-volume semiconductor manufacturing (See Methods for detailed evaluation conditions). For the overlay metrology in this study, more than 10 million overlay values were generated across a single wafer—representing a 1987-fold increase in data points compared to scanning electron microscopy (SEM). Each point was measured in 0.001 s, which is significantly faster than SEM and considered acceptable for practical use in semiconductor manufacturing (Table 3).

While IMMSE provides high-density metrology data with high throughput, improving throughput is essential in practical process monitoring. To address the practical need for faster wafer-level inspection in process monitoring applications, several strategies can be considered to optimize the throughput of the IMMSE system. First,

selective wavelength sampling can significantly reduce measurement time, focusing on a subset of wavelengths that are known to exhibit strong sensitivity to specific film parameters or structural variations. Second, the number of polarization conditions used for measurement can be adjusted based on the required sensitivity and complexity of the target parameter. For example, thin film thickness measurements often yield sufficient accuracy with four polarization states, and in some cases, monitoring trends over time may require only one or two states. However, it should be noted that certain applications such as overlay require $3 \times 3$ MM reconstruction, and thus cannot be performed with reduced polarization sets. Finally, the number of measured shots across the wafer can be reduced by selecting representative regions, similar to the strategy used in conventional point-based metrology tools. This is especially effective in production environments where wafer uniformity is relatively high and full-wafer coverage is not always necessary. Collectively, these optimization strategies allow the IMMSE system to be flexibly configured for high-throughput process monitoring, while retaining its capability for high-resolution, full-field measurements when required.

Previous studies have made significant advances in addressing specific optical imperfections in ellipsometry systems. These include the phase retardation of rotating compensators[51], the initial angular offsets of polarizers and compensators across wavelengths[52], nulling-based optimization approaches[53], and selective system calibration procedures for ellipsometry system, including polarizers and compensators[54] (Table 4).

**Table 2 | DRAM metrology Accuracy of the IMMSE**

| | Film thickness | | CD | | Overlay | |
|---|---|---|---|---|---|---|
| | Process 1 | Process 2 | Process 1 | Process 2 | Process 1 | Process 2 |
| Accuracy (nm) (Reg. Model RMSE) | 0.62 | 0.49 | 0.11 | 0.10 | 0.41 | 0.5 |

**Table 3 | Throughput of conventional SEM and IMMSE**

| | Measurement Points (pts) | Measurement Time (s) | Throughput (s/pts) |
|---|---|---|---|
| SEM | 5994 | 4196 | 0.700 |
| IMMSE | 11,912,908 | 12,600 | 0.001 |

The throughput is calculated based on the measurement points over full wafer area and total measurement time.

These works have primarily been demonstrated on standardized samples or through contrast-based imaging, and their calibration strategies have generally targeted specific components or error sources[51–55]. In this context, the IMMSE employs a generalized calibration framework that corrects signal distortions from all optical components in the illumination path, imaging optics, and polarizers by using the relative transmittance. This approach comprehensively accounts for the entire optical system, ensuring consistent and reliable measurements across the wide field of view and broad spectral range. This capability is particularly advantageous for accurately characterizing sub-nanometer structural variations in DRAM wafers.

While the IMMSE system was developed primarily for semiconductor metrology, its underlying optical and analytical principles make it applicable to a range of scientific and clinical domains that benefit from wide-field, high-throughput polarimetric measurements. In particular, the ability to acquire spatially dense MM data over a broad spectral range and large FOV opens possibilities in life sciences, medical diagnostics, and analytical chemistry.

In its current configuration, the IMMSE system employs linear polarization states for both generation and analysis, yielding a reduced $3 \times 3$ subset of the full MM. Despite this limitation, prior studies have shown that essential polarization parameters—such as linear diattenuation, linear retardance, and linear depolarization—can be reliably extracted from this subset[56,57]. These parameters have been applied in biomedical contexts, including the development of flexible fiber-based $3 \times 3$ MM probes for ex vivo tissue characterization[58], and in identifying optical signatures in retinal nerve fiber layers[59], visualizing anatomical features in brain tissue[60], and highlighting characteristic patterns in cancerous tissues[61].

These applications underscore the practical utility of partial MM measurements, especially in settings requiring compact instrumentation and simplified acquisition. Looking ahead, the IMMSE system's modular design allows for potential expansion beyond linear polarization. For instance, integrating broadband wide-field waveplates could enable full $4 \times 4$ MM acquisition, granting access to circular dichroism and comprehensive polarization characterization. Realizing this capability would require precise calibration of the wavelength- and position-dependent retardance of the waveplates, but the in-situ polarization correction framework presented in this work offers a promising foundation for addressing these challenges.

## Methods
### Optical setup
A broadband laser-driven light source (LDLS), covering a range of 170 nm to 2500 nm, was used as a light source of the system. A custom-designed monochromator was utilized to generate monochromatic light for spectral imaging, using the broadband light source. The wavelength range of the monochromator was from 250 nm–1100 nm, with a bandwidth of 3 nm full-width half-maximum (FWHM). The monochromator employed a Czerny-Turner configuration, which consists of rotating diffractions grating and two concave mirrors. To cover the broad wavelength range, three diffraction gratings, each covering a different wavelength range was employed. Using a high-speed direct drive type motor, we achieved a rapid switching time of 30 μs for a wavelength shift of 5 nm.

The output monochromatic light was delivered to the illumination optics using a multi-mode optical fiber and was incident on the sample at an angle of 65°. An engineered diffusing glass with the diverging angle of 5° FWHM was used to achieve Köhler illumination while maintaining a large illumination area. The converging angular spread of the illumination light was designed to be 3.2°. For the consistent angular properties over the field of view (FOV), a telecentric lens array with no magnification was configured.

The imaging module of the proposed IMMSE system was designed to achieve an ultra-wide FOV while maintaining high optical performance and minimizing chromatic aberration. The imaging optics adopted a multi-mirror reflective configuration, known as Offner configuration, which inherently minimizes chromatic dispersion and imaging distortion across a broad wavelength range. The Offner relay was configured with a numerical aperture (NA) of 0.06 and a telecentricity angle below 0.01°, balancing the trade-off between optical resolution and the system's sensitivity to nanoscale structural variations.

Although the imaging optics employed a reflective configuration, residual chromatic aberration was unavoidable due to the presence of the wire-grid polarizer and the image sensor's window glass. To minimize image distortion across the wide FOV, the curvature ratio between two concentric spherical mirrors of the Offner relay was precisely adjusted in design. By tuning the mirror curvatures, the image distortion was reduced to <0.19% across the entire FOV and the residual chromatic aberration was estimated as a focal plane shift of <40 μm across the system's full wavelength range.

Due to the oblique illumination geometry with a 65° incidence angle, the image plane is inherently tilted with respect to the sensor plane. Under the Scheimpflug condition, the required sensor tilt increases with magnification, which can significantly reduce light collection efficiency and complicate mechanical alignment. To avoid these issues, a 1× magnification was selected and minimized the necessary sensor tilt while preserving optical performance. The position and orientation of the image sensor were then precisely adjusted using a custom-designed 6-axis mechanical stage to satisfy the Scheimpflug condition. This ensured that the entire imaging plane was maintained within the system's depth of focus, thereby minimizing defocus-induced measurement errors.

The detector integrated into the system is a scientific CMOS (sCMOS) sensor featuring a resolution of 3200 × 3200 pixels with a pixel pitch of 6.5 μm, yielding an active sensing area of 20.8 mm × 20.8 mm. The sensor supports 16-bit analog-to-digital conversion, enabling 65,536 discrete intensity levels per pixel. The quantum efficiency of the sensor exceeds 30% in the short-wavelength region and reaches up to 95% within the visible spectrum, ensuring high photon-to-electron conversion efficiency across a broad spectral range. With a full well capacity of 15,000 electrons, conversion gain of 0.23 e−/count, and a low read noise of 1.6 e−, the system ensures high dynamic range and fine quantization accuracy. Active air cooling maintains the dark current at 1.27 e−/pixel/sec at 0 °C, supporting low-noise, high-speed imaging performance.

**Table 4 | Comparison of representative imaging Mueller matrix ellipsometry systems**

|  | This work | Jin et al.[51] | Chen et al.[52] | Braeuninger-Weimer et al.[53] | Chao et al.[54] |
|---|---|---|---|---|---|
| System Configuration | Dual rotating polarizer (3 × 3 MM) | PCrSA (Δ, ψ) | Dual rotating compensator (4 × 4 MM) | Fixed PCSA (Film model parameters) | Dual rotating compensator (4×4 MM) |
| Demonstrated Wavelength (nm) | 250–1100 | 650 | 400–700 | 390, 450, 550 | 400-700 |
| Field of View (mm²) | 20 × 20 | 5.7 × 4.3 | 4 × 4 | 2 × 2 | - |
| Residual Spectral Error | 0.004 | - | 0.01 | - | - |
| Inhomogeneity Correction Items | Comprehensive relative transmittance | Compensator | Polarizer and compensator | Polarizer and compensator | Polarizer, compensator, beam splitter, beam splitter, and objective lens |

Key features of the proposed system and four representative imaging Mueller matrix ellipsometry systems reported in the literature are compared in the table.

The proposed system utilized a dual rotating polarizer configuration to measure the $3 \times 3$ Mueller matrix (MM)[44]. A polarization state generator (PSG) controlled the polarization state of the illumination light, while a polarization state analyzer (PSA) characterized the polarization sensitivity of the sample wafer. The polarization state manipulation was achieved by rotating wire-grid polarizers glass with the size of 50 mm × 50 mm, mounted on hollow rotary motors.

The wire-grid polarizer is one of the most critical optical components in the IMMSE system. It was used to manipulate the polarization state of the illumination light and analyze the polarization sensitivity of the measured structures. To achieve this, we employed a high-performance wire-grid polarizer (Moxtek, US), which exhibits an exceptionally high extinction ratio. Notably, the polarizer maintains an extinction ratio >800 even at wavelengths below 300 nm, ensuring reliable polarization contrast in the short-wavelength region. In addition, it offers a transmittance level of ~62%, enabling efficient light throughput. These characteristics are essential for preserving the accuracy of polarization-resolved measurements across the FOV and a broad spectral range.

The alignment optics was designed to adjust both wafer position and image focus. It was integrated with a four-channel LED light source, including RGB and infrared, a 10 × magnification objective lens, laser autofocus module, and an image sensor.

## System factor determination

In this section, we propose the detailed information on the system factor determination, starting from Eq. (6) in "System factor analysis".

In order to determine the values of $H_{\alpha,i} = \sum_n \alpha_n K_{in}$ and $H_{\beta,j} = \sum_n \beta_n K_{nj}$, we revisited the distinctive property of the kernel factor

$$K_{ij} = \mathbf{q}_j^{\mathrm{T}} \mathbf{M} \mathbf{p}_i = \begin{bmatrix} 1 & \cos 2\theta_{\alpha,j} & \sin 2\theta_{\alpha,j} \end{bmatrix} \begin{bmatrix} 1 & m_{12} & m_{13} \\ m_{21} & m_{22} & m_{23} \\ m_{31} & m_{32} & m_{33} \end{bmatrix} \begin{bmatrix} 1 \\ \cos 2\theta_{\beta,i} \\ \sin 2\theta_{\beta,i} \end{bmatrix} \quad (8)$$

Leveraging this property, $H_{\alpha,i}$ and $H_{\beta,j}$ can be expressed as harmonic forms:

$$H_{\alpha,i} = \eta_\alpha \left( 1 + U_\alpha \cos 2\phi_i + V_\alpha \sin 2\phi_i \right) \quad (9)$$

$$H_{\beta,j} = \eta_\beta \left( 1 + U_\beta \cos 2\phi_j + V_\beta \sin 2\phi_j \right) \quad (10)$$

where $\eta_\alpha$ and $\eta_\beta$ are the scaling constants, $U_\beta$, $V_\beta$, $U_\alpha$, $V_\alpha$ are the harmonic parameters, $\phi_i$ and $\phi_j$ are the $i$-th and the $j$-th ideal rotation angle of the polarizers, evenly spaced over 360 degrees. In the Fourier domain, the unique property of the kernel factor is represented as exhibiting zero values except at the points corresponding to the reciprocal indices of −1,0 and 1, expressed as follows:

$$\sum_n R_{nj} H_{\alpha,n} \cos 2m\phi_n = \frac{1}{H_{\beta,j}} \sum_n K_{nj} \cos 2m\phi_n = 0, \text{ if } m = 2, 3, \ldots, \frac{N_\theta}{2} - 2 \quad (11)$$

$$\sum_n R_{nj} H_{\alpha,n} \sin 2m\phi_n = \frac{1}{H_{\beta,j}} \sum_n K_{nj} \sin 2m\phi_n = 0, \text{ if } m = 2, 3, \ldots, \frac{N_\theta}{2} - 2 \quad (12)$$

$$\sum_n R_{in} H_{\beta,n} \cos 2m\phi_n = \frac{1}{H_{\alpha,i}} \sum_n K_{in} \cos 2m\phi_n = 0, \text{ if } m = 2, 3, \ldots, \frac{N_\theta}{2} - 2 \quad (13)$$

$$\sum_n R_{in} H_{\beta,n} \sin 2m\phi_n = \frac{1}{H_{\alpha,i}} \sum_n K_{in} \sin 2m\phi_n = 0, \text{ if } m = 2, 3, \ldots, \frac{N_\theta}{2} - 2 \quad (14)$$

where $N_\theta$ is total number of rotations. Then, $U_\beta$, $V_\beta$, $U_\alpha$, $V_\alpha$ can be obtained by solving following systems of equations using least squares method.

$$U_\alpha \sum_n R_{nj} \cos 2\phi_n \cos 2m\phi_n + V_\alpha \sum_n R_{nj} \sin 2\phi_n \cos 2m\phi_n = -\sum_n R_{nj} \cos 2m\phi_n \quad (15)$$

$$U_\alpha \sum_n R_{nj} \cos 2\phi_n \sin 2m\phi_n + V_\alpha \sum_n R_{nj} \sin 2\phi_n \sin 2m\phi_n = -\sum_n R_{nj} \sin 2m\phi_n \quad (16)$$

$$U_\beta \sum_n R_{in} \cos 2\phi_n \cos 2m\phi_n + V_\beta \sum_n R_{in} \sin 2\phi_n \cos 2m\phi_n = -\sum_n R_{in} \cos 2m\phi_n \quad (17)$$

$$U_\beta \sum_n R_{in} \cos 2\phi_n \sin 2m\phi_n + V_\beta \sum_n R_{in} \sin 2\phi_n \sin 2m\phi_n = -\sum_n R_{in} \sin 2m\phi_n \quad (18)$$

where $m = 2, 3, \ldots, \frac{N}{2} - 2$. After obtaining the harmonic parameters, we can calculate the kernel factor using following equation:

$$K_{ij} = R_{ij} H_{\alpha,i} H_{\beta,j} = \eta R_{ij} \left( 1 + U_\alpha \cos 2\phi_i + V_\alpha \sin 2\phi_i \right) \left( 1 + U_\beta \cos 2\phi_j + V_\beta \sin 2\phi_j \right) \quad (19)$$

where the scaling constant $\eta = \eta_\alpha \eta_\beta$ can be determined based on the condition that the average value of the kernel factor is 1. As a result, we can determine both system factors $\alpha_j$ and $\beta_i$ as follows,

$$\alpha_j = \frac{1}{\eta_\beta} \cdot \frac{\sum_n L_{nj}}{1 + U_\beta \cos 2\phi_j + V_\beta \sin 2\phi_j} \quad (20)$$

$$\beta_i = \frac{1}{\eta_\alpha} \cdot \frac{\sum_n L_{in}}{1 + U_\alpha \cos 2\phi_i + V_\alpha \sin 2\phi_i} \quad (21)$$

## Mueller matrix calculation

After determining the system factors, kernel factor can be directly obtained by removing system factors from the measured intensity, based on the Eq. (5). Since the kernel factor was defined as Eq. (8), which implies the linear combination of orthogonal bases associated with each MM element, we can calculate all MM components using the following relationships:

$$m_{11} = 1 \text{ (definition)} \tag{22}$$

$$m_{12} = \frac{2}{N_\theta} \sum_{t=1}^{N_\theta} K_{5t,3t} \cos\theta_{\alpha,3t} \tag{23}$$

$$m_{13} = \frac{2}{N_\theta} \sum_{t=1}^{N_\theta} K_{5t,3t} \sin\theta_{\alpha,3t} \tag{24}$$

$$m_{21} = \frac{2}{N_\theta} \sum_{t=1}^{N_\theta} K_{5t,3t} \cos\theta_{\beta,5t} \tag{25}$$

$$m_{22} = \frac{4}{N_\theta} \sum_{t=1}^{N_\theta} K_{5t,3t} \cos\theta_{\alpha,3t} \cos\theta_{\beta,5t} \tag{26}$$

$$m_{23} = \frac{4}{N_\theta} \sum_{t=1}^{N_\theta} K_{5t,3t} \sin\theta_{\alpha,3t} \cos\theta_{\beta,5t} \tag{27}$$

$$m_{31} = \frac{2}{N_\theta} \sum_{t=1}^{N_\theta} K_{5t,3t} \sin\theta_{\beta,5t} \tag{28}$$

$$m_{32} = \frac{4}{N_\theta} \sum_{t=1}^{N_\theta} K_{5t,3t} \cos\theta_{\alpha,3t} \sin\theta_{\beta,5t} \tag{29}$$

$$m_{33} = \frac{4}{N_\theta} \sum_{t=1}^{N_\theta} K_{5t,3t} \sin\theta_{\alpha,3t} \sin\theta_{\beta,5t} \tag{30}$$

Since we employed the combinations of polarization states that the PSG and PSA are rotated with 5:3 rotation angle ratio, $K_{5t,3t}$ represents the kernel factor at the $5t$-th PSG angles and the $3t$-th PSA angles in rotations.

## Data acquisition procedure

Wafer-level data acquisition is performed using a stepping-stage approach, where the system sequentially measures individual shot regions. Each shot is divided into four quadrants ($2 \times 2$ layout) due to the $20 \times 20$ mm FOV of the imaging optics, whereas typical shot sizes range from 25–35 mm. The stage steps across quadrants within a shot, and moves to the next shot with alignment based on shot mark recognition.

At each quadrant, a MM spectroscopic image is captured. Spectral data are not extracted from the entire image but rather from pre-defined target positions such as DRAM cell blocks. Each target is covered by one of the quadrants. To extract the data, a small neighborhood (e.g., $15 \times 15$ pixels) pixels around each target are selected, and subpixel interpolation is applied before averaging. The resulting spectra from each quadrant are then aggregated to form the complete dataset for the shot.

Since the system acquires spectral data directly from target positions, image-level fusion between quadrants is not performed, and spatial consistency is ensured through calibrated stage control.

A typical measurement recipe involves 9 PSG/PSA combinations. For each condition, the PSG and PSA rotate simultaneously and stabilize within around 0.3 s. A spectral scan across 35 wavelengths (270-780 nm) is then performed, taking about 2.5 s on average including monochromator movement and image acquisition. Each polarization condition therefore requires about 2.8 s, yielding roughly 25 s per quadrant. Stepping between quadrants takes about 1.2 s per move, and shot-to-shot alignment takes about 6.2 s. As a result, one full shot requires about 110 s to measure. For a full wafer, which generally contains around 100−120 shots, the total measurement time amounts to ~3.1−3.7 h.

## Machine learning for spatially dense metrology

For the spatially dense metrology across the entire wafer area, we employed a machine learning algorithm. Ridge regression, a regularized variant of least-squares regression, mitigates overfitting by incorporating a penalty term proportional to the squared magnitude of the coefficients. For model training, reference metrology data−such as SEM measurements−were collected from specific locations on the wafer. At each reference location, corresponding $3 \times 3$ MM spectra were obtained using the IMMSE system by averaging over $15 \times 15$ pixel regions centered on the nearest cell block.

The dataset consisting of paired MM spectra and reference values was partitioned into three subsets: training, validation, and test sets. This partitioning enabled proper optimization of the regression model, including the selection of an appropriate regularization constant. Once trained, the model was applied to the MM spectra acquired across the entire wafer, enabling spatially dense prediction of metrology data with >10 million measurement points across the wafer.

For the thin layer measurement described in "DRAM metrology and applications", the reference dataset was limited to only 49 points. To address this, we repeated the data splitting process 1000 times, keeping the test set size at 30% to determine the optimal data split configuration. We exclusively used the $m_{33}/m_{22}$ spectrum, corresponding to the C-spectrum in conventional ellipsometry45, due to the isotropic nature of the thin film layer, where the off-diagonal MM components are expected to be zero. Despite using the single MM spectrum, the dataset size remains small, consisting of 35 wavelengths and 17 training samples. To address this limitation, we employed principal component analysis (PCA) and reduced the dimensionality of the dataset's features from 35 to 16. The optimized regularization constant of the regression model for the wafer sample processed under the Process 1 was determined to be 0.00018, yielding an R2 score of 0.21 and an RMSE of 0.62 nm on the test set. Although this result showed a low R2 value, it was sufficient for identifying the unique thickness variation trends across the entire wafer area. For the wafer sample processed under the Process 2, the regression model was optimized with a regularization constant of 0.000035, achieving an R2 score of 0.95, and an RMSE of 0.49 nm on the test set.

In the measurement of the DRAM capacitor described in "DRAM metrology and applications"", the reference CD dataset consisted of 536 points, with 20% allocated to the test set. The remaining data was used to determine an appropriate regularization constant through 4-fold cross-validation. A total of 280 features, derived from 35 wavelengths and 8 MM components, were utilized to train the regression model. For the regression model trained on the wafer processed under the Process 1, the optimized regularization constant was 0.015, with an R2 score of 0.715 and an RMSE of 0.106 nm on the test set. The regression model for the sample processed through the Process 2 was optimized with a regularization constant of 0.021, yielding an R2 score of 0.589 and an RMSE of 0.103 nm on the test set.

The data splitting for the overlay measurement in the same section was performed under the same conditions with the CD measurement. For the regression model trained on the wafer processed under the Process 1, the optimized regularization constant was 0.00006, with an R2 score of 0.74 and an RMSE of 0.41 nm on the test set. The regression model for the sample processed through the Process 2 was optimized with a regularization constant of 0.0009, yielding an R2 score of 0.50 and an RMSE of 0.36 nm on the test set.

Although other models, such as neural networks, were also considered, Ridge regression was selected due to its favorable trade-off between accuracy, computational efficiency, and interpretability. This choice was particularly suitable for the overlay measurement in Process 1 and 2 of Fig. 6, where the relationship between the MM spectra and the garget parameter appeared to be locally linear.

### Overlay measurement repeatability

To verify the repeatability of the IMMSE system, the overlay between the bit line (BL) and active (ACT) patterns on a DRAM sample wafer was measured. A total of seven repeated measurements were conducted at the same wafer over a period of ~200 h, under identical system and environmental conditions. The system-induced signal distortion of each repeated measurement was corrected using the system factors defined before the initial measurement, to verify that the system factors remain robust under environmental fluctuations.

The regression model was trained using identical reference overlay values and the IMMSE spectra obtained individually from each repeated measurement. For the repeatability analysis, the standard deviation of repeated measurements was first calculated at each of the 12,672 measurement points within a single exposure shot located at the wafer center. The 3-sigma repeatability was then obtained by averaging these standard deviations across all measurement points in the shot area. The resulting repeatability was 0.36 nm, demonstrating that the IMMSE system maintains highly stable performance for sub-nanometer process control and monitoring in semiconductor manufacturing over extended periods.

### Data availability

The raw and processed datasets generated in this study are subject to the company's confidentiality policy, which generally prohibit external release. Researchers may contact the corresponding authors to request access, and such requests will be considered on a case-by-case basis for non-commercial research purposes. Requests will be responded to within 4 weeks of receipt. Alternative colour versions of certain figures are available upon request.

### Code availability

The code used in this study is subject to the company's confidentiality policy and contractual obligations, which generally prohibit external release. Researchers may contact the corresponding authors to request access, and such requests will be considered on a case-by-case basis for non-commercial research purposes. Requests will be responded to within 4 weeks of receipt.

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

## Acknowledgements

This research was supported by the Semiconductor R&D Center, Samsung Electronics Co., Ltd.

## Author contributions

J.O., J.S. (Jaehyeon Son). and C.Y. conceived the original idea and developed the system and theory. E.H., J.A. (Jinwoo Ahn), J.L. (Jaewon Lee), J.O., J.S. (Jiyong Shin) and M.L. designed and integrated IMMSE system. J.L. (Jaewon Lee and Jinsoo Lee) developed the system software. J.S. (Jaehyeon Son) developed the system-induced signal distortion correction algorithm. J.O., J.S. (Jaehyeon Son), D.L., S.L. and C.Y. conducted experiment and analyzed the data. J.O., J.S. (Jaehyeon Son), C.Y. and E.H. wrote the manuscript with support from Y.S., J.A. (Jeongho Ahn), T.J., S.H. and M.L. J.O. and T.J. were in charge of the overall direction and planning. T.J. supervised the project. All authors provided critical feedback and helped shape the research.

## Competing interests

The authors declare no competing interests.
