## [Transparent Peer Review file · Nature Communications]

Ultra-Wide-Field Imaging Mueller Matrix Spectroscopic Ellipsometry for Semiconductor Metrology

Corresponding Author: Dr Taeyong Jo

Version 0:

Reviewer comments:

Reviewer #1

(Remarks to the Author)

Overlay error metrology is an important indicator for lithography quality evaluation. The manuscript proposed an IMMSE system for semiconductor metrology, which achieves large-area measurements with an impressive 20 mm × 20 mm field of view (FOV). Such a large FOV enables the potential full chip scanning in IC manuscript. The contents fall into the scope of Nature Communication and may have significant impact to both the research and industrial community of IC manufacturing. To be publishable, the authors should address the following weak points of their manuscript.

1. Since the most important feature of the proposed system is achieving the large FOV, more information of the imaging module should be provided to reveal the mechanism, principle and design. For example, the configurations and parameters of the imaging optics are necessary to analyze the quality of images. The specification of the detecting sensor is necessary to check the image resolution.
 2. And the readers may also want to know how the aberration and polarization distortion induced by the imaging module may affect the measurements, and whether the authors applied corrections. In the manuscript, it seems that only the effects induced by two polarizers are considered.
 3. How is the data on the entire wafer achieved, by scanning or stepping the stage? I'm also curious about how to fuse the images of each shot. I'm also wondering the time for each measurement step because each step includes the time for PSA and PSG rotating and wavelength shift. The current total time of 3.5 hours and the averaging time for one point is not sufficient to describe the efficiency of the proposed method.
 4. The title of Section 3 is discussion. Actually, it is a summary of method proposed in this manuscript, which had been already provided in other sections. In the discussion section, I expect to see the discussion on the results, the limits of the proposed method, as well as the comparison of results by using different data processing, etc.
- All in all, the method and results presented in the manuscript is very interesting and sound solid. however, it needs a major revision to become publishable.

Reviewer #2

(Remarks to the Author)

The authors proposed an ultra-wide-field imaging Mueller matrix spectroscopic ellipsometry (IMMSE) system for semiconductor metrology. The demonstrated IMMSE system achieves a 20 mm × 20 mm field of view (FOV) with 6.5 μm resolution. The system leverages a novel signal correction algorithm to ensure spectral consistency across the FOV and combines hyperspectral imaging with machine learning to enable spatially dense metrology. Some key innovations of this manuscript include 12,800× increase in data density and 580× faster throughput. This work addresses critical challenges in semiconductor manufacturing, such as detecting in-chip variations and optimizing processes like CVD and photolithography, and will be useful and inspiring for the relevant field. This reviewer would recommend the publication of this manuscript provided that the authors can make necessary revision and properly address the following comments and concerns.

1. The manuscript uses Ridge linear regression for the dense metrology. How does the Ridge linear regression compare with other machine learning methods, such as neural network?
2. Please comment on the system stability, such as thermal drift, vibration sensitivity, etc.
3. How does the measurement accuracy (or RMSE) compare with the industrial standard, e.g., the ITRS roadmap? Please

comment in the applicability of the proposed method in the current and future technology nodes.

4. In Table 3, the authors compared the throughput of their IMMSE system with SEM. IMMSE took 3.5 hours to complete the measurement. Although IMMSE measurement contains much more data point, in reality, it often is more important to complete the wafer inspection faster for overall throughput particularly when the measurement is used for process monitoring. Can the authors comment on how to further improve the total measurement time? For example, can one reduce the measurement point to get much shorter measurement time?

5. It might be better if a flowchart of data processing and the overall algorithm schematic can be included to clarify these various aspects.

Reviewer #3

(Remarks to the Author)

This paper proposes an ultra-wide imaging Mueller matrix ellipsometer for semiconductor metrology. By achieving a 20mm×20mm FOV with a spatial resolution of 6.5μm, they can obtain more than 10 million 3×3 Mueller matrix spectra to implement spatially dense semiconductor metrology. The authors have demonstrated impressive results in identifying spatial variations of the DRAM overlay to potentially improve the semiconductor manufacturing yield. However, after carefully reviewing this manuscript, in my opinion, this paper is only an incremental innovation in technique, and lacks original novelty in theory and method. The imaging Mueller matrix ellipsometry is a wide-spread and well-published technique, and its applications in semiconductor metrology for thin films, nanostructures, even overlays are also well published. Machine learning assisted data analysis for ellipsometry especially for complex multilayer thin films and nanostructures has also been well studied during recent years. The authors should clarify and strengthen their original novelty compared with published techniques. I cannot recommend the current manuscript for the publications in Nature Communication.

This paper proposes some detailed techniques for their configuration of imaging MMSE, such as the calibration of system-induced signal distortion, and the residual polarization in light source due to the rotating polarizer in the PSG and polarization sensitivity of the detector due to the rotating polarizer in the PSA by using so-called system factor analysis. I think it can be published in a specific journal in fields of optics or instruments. But more details about their configurations and techniques for how to achieve such an ultra-wide FOV with a relatively decent resolution should be provided.

Reviewer #4

(Remarks to the Author)

In this work, the authors present an innovative two-dimensional spectral ellipsometry system with a wide field of view and high resolution, particularly suitable for semiconductor metrology.

The system is capable of measuring 9 out of the 16 elements of the Mueller matrix of a material, but not those related to the interaction with the circular component of the Stokes vectors.

The article is well written, and the bibliography is comprehensive.

The strengths of the presented metrology apparatus are its wide field of view, high spatial density, and impressive throughput speed. In fact, the IMMSE covers an area of 20 mm x 20 mm with a resolution of 6.5 μm, providing for each pixel the spectral information of the Mueller matrix elements in the 270 nm – 780 nm range, with 15 nm sampling steps and a spectral resolution of 3 nm.

The system is also equipped with advanced signal correction and self-learning techniques.

All of this is impressive, but I believe the apparatus is significantly limited by the lack of any information related to circular polarization and circular dichroism. This not only excludes the possibility of studying an entire class of anisotropic materials, but I also believe the instrument may struggle in the analysis of thin multilayer structures, as it would not be possible to separate information about the thickness of each layer from that of its dielectric function.

For the same reason, this polarimeter cannot directly measure the depolarization of light and therefore cannot assess surface roughness. How, then, can it provide information on this important property of thin films?

Why did the authors not choose a rotating compensator-based solution, such as the one described in reference 19?

The authors should add the following information to the manuscript:

1. The brand of the polarizers used, or at least their extinction ratio.
2. The resolution of the analog-to-digital (A/D) conversion of the images acquired by the sCMOS image sensor used in the IMMSE system.

In my opinion, this work is undoubtedly of high scientific quality and quite innovative. However, I do not believe it has broad enough scientific relevance to justify publication in Nature Communications.

There are numerous applications of polarimetry in the life sciences and chemistry (e.g., detection of food adulteration, chemical and biological analysis, etc.), and others have been suggested (e.g., identification of metastatic tissues and viruses). However, in all these cases, it is essential to measure quantities such as the degree of polarization and dichroism, capabilities that, in my view, the IMMSE system in its current optical configuration cannot provide.

Version 1:

Reviewer comments:

Reviewer #1

(Remarks to the Author)

I think the authors made a good revision and addressed all my concerns on the manuscript. I think it can be accepted now. I just have only one optional suggestion that whether the authors can provide the repeatability at the same location by the proposed instrument and method, which is another critical index in volume IC manufacturing.

Reviewer #2

(Remarks to the Author)

The authors have addressed the comments and questions from this reviewer in a satisfactory way, therefore the revised manuscript is recommended to be accepted for publication.

Reviewer #3

(Remarks to the Author)

I thanks the authors for their efforts to response my concerns. The comparison of representative imaging Mueller matrix ellipsometry systems (Table 3. in the response to my comments) and relative comments on the corresponding references (also one more ref. is recommended Opt. Express 29: 32712-32727, 2021) should be added in the manuscript to clearly talk about the state-of-art of IMMSE and the contribution of this work to the community.

Reviewer #4

(Remarks to the Author)

I sincerely thank the authors for significantly improving the manuscript and answering my questions in a more than satisfactory manner. In its current form, I believe that the manuscript is certainly suitable for publication in Nature Communications.

Reviewer #1 (Remarks to the Author):

1. Since the most important feature of the proposed system is achieving the large FOV, more information of the imaging module should be provided to reveal the mechanism, principle and design. For example, the configurations and parameters of the imaging optics are necessary to analyze the quality of images. The specification of the detecting sensor is necessary to check the image resolution.

We appreciate the reviewer's valuable comment regarding the need for more detailed information about the imaging module, particularly in relation to the large field of view (FOV), which is a key feature of our proposed system. To address this concern, we have added detailed descriptions of both the imaging optics and the image sensor in the revised manuscript.

Specifically, for the imaging optics, we now include the full optical configuration based on a multi-mirror reflective system designed to achieve an ultra-wide FOV while minimizing aberrations. In addition, the revised manuscript explicitly describes how the Scheimpflug condition was incorporated into the optical design to compensate for the oblique illumination geometry, which had not been fully addressed in the original version.

Whereas the original manuscript only provided the FOV size (20 mm × 20mm), wavelength range (250 nm to 1100 nm), and numerical aperture (NA=0.06), the revised version additionally includes key optical parameters such as magnification (1×), telecentricity (0.01°), and inherent aberrations.

In addition, we provide comprehensive specifications of the scientific CMOS (sCMOS) image sensor, including its resolution (3200 × 3200 pixels), pixel pitch (6.5 μm), active sensor area (20.8 mm × 20.8 mm), and 16-bit analog-to-digital (A/D) conversion capability. The sensor's full well capacity (15,000 e⁻), read noise (1.6 e⁻), and conversion gain (0.23 e⁻/count) are also provided, along with the active air cooling mechanism that suppresses dark current to 1.27 e⁻/pixel/sec at 0 °C. Moreover, the sensor exhibits a quantum efficiency of at least 30% in the short-wavelength region and up to 95% within the visible spectrum, ensuring high photon-to-electron conversion efficiency across a broad spectral range. These specifications collectively support high dynamic range, low noise, and fine intensity quantization, which are critical for capturing spatial and spectral variations with high sensitivity across the large field of view (FOV).

The newly added information can be found in Section 4.1 (Optical setup) of the revised manuscript.

Line 515:

An Offner configuration, comprising a pair of concentric spherical mirrors, was employed as an imaging optics to minimize image distortion and chromatic aberration⁴²⁻⁴⁴ across the wide FOV of 20 mm × 20 mm and a broadband wavelength range of 250 nm to 1100 nm. The numerical aperture (NA) of the imaging optics was designed to be 0.06, considering the trade-off between the overall size of the optical system and its spectral sensitivity. A back-illuminated scientific complementary metal oxide semiconductor (sCMOS) image sensor was used to record signals reflected from the sample wafer. The position of the image sensor was precisely adjusted using a custom-designed 6-axis mechanical stage to satisfy the Scheimpflug imaging condition for the oblique incident angle of 65°.

Line 523:

The imaging module of the proposed IMMSE system was designed to achieve an ultra-wide FOV while maintaining high optical performance and minimizing chromatic aberration. The imaging optics adopted a multi-mirror reflective configuration, known as Offner configuration, which inherently minimizes chromatic dispersion and imaging distortion across a broad wavelength range. The Offner relay was configured with a numerical aperture (NA) of 0.06 and a telecentricity angle below 0.01°, balancing the trade-off between optical resolution and the system's sensitivity to nanoscale structural variations.

Although the imaging optics employed a reflective configuration, residual chromatic aberration was unavoidable due to the presence of the wire-grid polarizer and the image sensor's window glass. To minimize image distortion across the wide FOV, the curvature ratio between two concentric spherical mirrors of the Offner relay was precisely adjusted in design. By tuning the mirror curvatures, the image distortion was reduced to less than 0.19% across the entire FOV and the residual chromatic aberration was estimated as a focal plane shift of less than 40 μm across the system's full wavelength range.

Due to the oblique illumination geometry with a 65° incidence angle, the image plane is inherently tilted with respect to the sensor plane. Under the Scheimpflug condition, the required sensor tilt increases with magnification, which can significantly reduce light collection efficiency and complicate mechanical alignment. To avoid these issues, a $1\times$ magnification was selected and minimized the necessary sensor tilt while preserving optical performance. The position and orientation of the image sensor were then precisely adjusted using a custom-designed 6-axis mechanical stage to satisfy the Scheimpflug condition. This ensured that the entire imaging plane was maintained within the system's depth of focus, thereby minimizing defocus-induced measurement errors.

The detector integrated into the system is a scientific CMOS (sCMOS) sensor featuring a resolution of 3200×3200 pixels with a pixel pitch of $6.5 \mu\text{m}$, yielding an active sensing area of $20.8 \text{ mm} \times 20.8 \text{ mm}$. The sensor supports 16-bit analog-to-digital conversion, enabling 65,536 discrete intensity levels per pixel. The quantum efficiency of the sensor exceeds 30% in the short-wavelength region and reaches up to 95% within the visible spectrum, ensuring high photon-to-electron conversion efficiency across a broad spectral range. With a full well capacity of 15,000 electrons, conversion gain of $0.23 \text{ e}^-/\text{count}$, and a low read noise of 1.6 e^- , the system ensures high dynamic range and fine quantization accuracy. Active air cooling maintains the dark current at $1.27 \text{ e}^-/\text{pixel}/\text{sec}$ at 0°C , supporting low-noise, high-speed imaging performance.

2. And the readers may also want to know how the aberration and polarization distortion induced by the imaging module may affect the measurements, and whether the authors applied corrections. In the manuscript, it seems that only the effects induced by two polarizers are considered.

We sincerely thank the reviewer for this insightful comment. The imaging module plays a crucial role in the system's polarization response and spatial fidelity, and we agree that this aspect required a more rigorous explanation.

First, regarding polarization distortion: In our system model (Eq. 2: $L_{ij} = \vec{d}^T Q_j M P_i \vec{s}$), the vector \vec{d} represents the cumulative polarization sensitivity of all detection-side components, including Offner relay mirrors, camera window, and the image sensor. Each component is modeled using a Mueller matrix, and their combined effect is fully incorporated in the system-level expression given in Eq. 1 (Eq. 1: $\vec{s}_{\text{out}} = M_1 M_2 \cdots M_n \vec{s}_{\text{in}}$). Hence, the measured intensity reflects not only the influence of the polarizers (PSG and PSA), but also any field-dependent polarization effects introduced by the imaging optics and sensor.

However, we acknowledge that the original description of \vec{d} in section 2.3 – as “the polarization sensitivity of the imaging optics and sensor” – may have been overly simplistic. We have revised the text in Section 2.3 to clarify that \vec{d} captures the net Mueller matrix of all optical elements following the PSA, thereby addressing spatially varying polarization distortions within the detection path.

Second, regarding optical aberrations: Our system covers a broad spectral range (250 nm – 1100 nm), which inherently poses a challenge for chromatic aberration. To mitigate this, we employ a fully reflective imaging system. The Offner relay configuration was specifically chosen because it effectively suppresses both chromatic and geometric aberrations while maintaining a wide field of view (20 mm \times 20 mm).

Geometric aberrations, including coma, astigmatism, and field curvature, can significantly impact image geometry, even at subtle levels. This is particularly concerning in semiconductor metrology, where even minor spatial errors can lead to inaccurate spectral interpretation. Although the Offner relay is designed to minimize geometric aberrations, residual geometric distortion still remains. To correct this, we implemented a calibration-based post-processing step: a known reference pattern is scanned using the motorized stage, and the deviation from ideal positions is used to construct a spatial distortion map. This map is applied after polarization demodulation to ensure spatial integrity of the measurement.

Additionally, we observed that the image sensor exhibits residual nonlinearity, especially in low-intensity regions, despite the manufacturer's built-in correction. This is particularly important in our system due to the large intensity range across different polarization states. We therefore performed an extra linearity calibration using a bare Si wafer under 500 nm illumination, capturing exposure-dependent responses and applying spline-based correction. This refinement was incorporated into our preprocessing pipeline prior to Mueller matrix reconstruction.

These considerations are now reflected in the revised version of Section 2.2. In particular, the section has been updated to emphasize how both polarization distortion and geometric aberration affect measurement fidelity, and how our system compensates for both via careful optical design and post-processing correction.

Line 118:

2.2. System-induced signal distortion

One of the key features of the proposed wide-field IMMSE system for semiconductor manufacturing is its ability to deliver consistent and reliable measurement results over the FOV. Specifically, measurements taken at the same point on a sample should remain consistent, whether the point is located at the center or the edge of the FOV. However, in practice, signal distortion occurs depending on the positions within the FOV.

The measurement discrepancies within the large FOV are expected to arise from the non-uniformity of optical performance over their operational regions, such as the transmittance or reflectance of lenses, mirrors and polarizers. Specifically, the signal recorded at each pixel is the result of light propagated through different regions of optical components. These components exhibit position- and wavelength-dependent variations in transmittance and reflectance, which leads to pixel-wise discrepancies in the measured intensity. In addition to the performance inconsistency of the optical components, the signal also changes with the rotation of the polarizer, as the light passes through different areas of polarizer by the rotation, as shown in Fig. 2a.

Previous research on the imaging ellipsometry addressed the inhomogeneity of retardation in rotating compensator by employing an extra phase constant in the equation^{34,35}. In this work, we introduce a novel method to obtain reliable signals across the FOV by eliminating system-induced signal distortion in the dual rotating polarizer configuration. This was achieved by employing condition-specific relative transmittance (RT) constant and compensating for the performance changes in the PSG, PSA, and wavelength.

In addition to polarization-induced intensity variations, residual geometric distortion from the imaging optics can degrade the spatial alignment of hyperspectral data across the wide FOV. Although the Offner-type reflective relay used in the imaging module minimizes chromatic and geometric aberrations, slight spatial distortion remains due to practical implementation factors. To address this, we applied a geometric distortion correction during post-processing. A reference target was used to measure the pixel-wise displacement vectors across the field, and these vectors were used to remap each image to a common spatial grid. In semiconductor metrology, even a few micrometers of spatial misalignment can affect spectral accuracy, especially when analyzing pixel-wise variations across polarization conditions. Our correction ensures spatial consistency across wavelengths and polarization states, supporting accurate Mueller matrix reconstruction at high spatial resolution.

Furthermore, while the camera manufacturer provides basic linearity correction, some residual nonlinear response remains, particularly in low-intensity regions. This is critical in our system due to large intensity variation across polarization states. Additional calibration was performed using a bare Si wafer under 500 nm illumination, with the exposure time incrementally increased by 0.015 ms. A separate dark measurement was also taken. The resulting intensity curve was corrected using spline interpolation to generate a refined response map.

Line 151:

The intensity and partial polarization characteristics of light are described using the Stokes vector. The relationship between the light before and after interacting with the sample can be expressed as follows:

$$\vec{s}_{\text{out}} = M_1 M_2 \cdots M_n \vec{s}_{\text{in}} \quad (1)$$

where \vec{s}_{in} and \vec{s}_{out} are Stokes vectors of input and output light, respectively, M_1, M_2, \dots, M_n are the MMs of optical components in the system. Since the dual rotating polarizer configuration was employed in this study, the MM is represented as a 3×3 square matrix, excluding circular polarization characteristics⁴⁵. Based on Eq. (1), the intensity measured using the proposed system can be expressed as a matrix product of MMs of the polarization manipulation components and the sample wafer, as shown below:

$$L_{ij} = \vec{d}^T Q_j M P_i \vec{s} \quad (2)$$

where L_{ij} is the measured intensity at the i -th PSG condition and the j -th PSA condition. M represents the MM of the sample wafer, P_i and Q_j denote the MMs of the PSG and the PSA, respectively. \vec{s} is Stokes vector of illumination light and \vec{d} represents the cumulative polarization sensitivity of all detection-side components after the PSA. This includes the Offner relay, sensor window, and the sCMOS image sensor, all modeled as Mueller matrices. To incorporate the RT of the optical system, we reformulated both polarizer MMs P_i and Q_j as follows:

3. How is the data on the entire wafer achieved, by scanning or stepping the stage? I'm also curious about how to fuse the images of each shot. I'm also wondering the time for each measurement step because each step includes the time for PSA and PSG rotating and wavelength shift. The current total time of 3.5 hours and the averaging time for one point is not sufficient to describe the efficiency of the proposed method.

We thank the reviewer for these insightful questions regarding the data acquisition process and measurement timing. Below, we provide a detailed explanation of how wafer-scale data is obtained, how measurements are organized and combined, how the total time is computed, and how the efficiency of the proposed method is related.

1. Data acquisition method:

The system measures the wafer using a stepping-stage approach, where each shot region is sequentially aligned and measured. Due to the imaging field of view (FOV) of 20×20 mm, while typical shot sizes are 25-35 mm, each shot is divided into four quadrants (2x2 layout). The stage steps between quadrants to cover the entire shot, and then moves to the next shot.

At each quadrant, the system captures a Mueller matrix spectroscopic image, where each pixel contains full spectral information. Instead of measuring the entire field, we focus on predefined target positions (e.g., DRAM cell blocks), which are distributed across the shot. Each target is covered by one of the four quadrants.

To extract spectral data at each target, we select a small neighborhood (e.g., 15×15 pixels) around the target location and apply subpixel interpolation followed by averaging. These local spectra are then aggregated across the four quadrants to reconstruct the complete dataset for the shot.

Instead of fusing the images of each quadrant, we directly extract and combine spectral data from the relevant regions. Therefore, image-level stitching is not performed.

2. Detailed measurement timing and discussion on efficiency

In a typical measurement configuration (recipe), spectroscopic image acquisition is performed for 9 predefined PSG/PSA angle combinations at each quadrant. For each polarization condition, the PSG and PSA rotate simultaneously and stabilize within approximately 0.3 seconds. A full spectral scan covering 35 wavelengths from 270 nm to 780 nm is then carried out, taking around 2.6 seconds on average. Consequently, each polarization condition takes approximately 2.9 seconds, resulting in a total measurement time of roughly 26 seconds per quadrant.

Stage movement is also involved in the acquisition process. Transitions between quadrants within the same shot require about 1.2 seconds each, and three such movements are needed per shot. Moving to the next shot involves shot mark (or key pattern) alignment, which typically takes about 6.2 seconds.

Combining all steps, the total time to complete the measurement for one shot is approximately 114 seconds. For a full wafer, which generally contains around 110 shots, the total measurement time amounts to approximately 3.5 hours. This estimates includes all polarization and spectral measurements, stage transitions, shot alignment.

It is also worth noting that the total measurement time is determined primarily by the number of shot regions rather than the number of target points, as all predefined locations within each shot are measured together during the quadrant-wise acquisition process.

While the proposed method requires several hours of a full-wafer scan, achieving a similar spatial sampling density using point-based instruments would be virtually infeasible in terms of total measurement time. This highlights the practical advantage of our imaging-based approach for high-resolution, large-area mapping.

Moreover, we have observed that certain process-related variations or patterns only become apparent when measurements are performed at this level of spatial granularity – these are often missed with sparse or limited-point measurements.

These clarifications have been incorporated into the revised manuscript. We have added a detailed explanation of the stepping-based wafer scanning procedure, including the quadrant-wise acquisition structure, measurement timing (covering polarization adjustment, monochromator scan, and stage movement).

Line 598:

4.4. Data acquisition procedure

Wafer-level data acquisition is performed using a stepping-stage approach, where the system sequentially measures individual shot regions. Each shot is divided into four quadrants (2×2 layout) due to the 20 × 20 mm FOV of the imaging optics, whereas typical shot sizes range from 25 to 35 mm. The stage steps across quadrants within a shot, and moves to the next shot with alignment based on shot mark recognition.

At each quadrant, a Mueller matrix spectroscopic image is captured. Spectral data are not extracted from the entire image but rather from predefined target positions such as DRAM cell blocks. Each target is covered by one of the quadrants. To extract the data, a small neighborhood (e.g., 15 × 15 pixels) pixels around each target are selected, and subpixel interpolation is applied before averaging. The resulting spectra from each quadrant are then aggregated to form the complete dataset for the shot.

Since the system acquires spectral data directly from target positions, image-level fusion between quadrants is not performed, and spatial consistency is ensured through calibrated stage control.

A typical measurement recipe involves 9 PSG/PSA combinations. For each condition, the PSG and PSA rotate simultaneously and stabilize within around 0.3 s. A spectral scan across 35 wavelengths (270-780 nm) is then performed, taking about 2.5 s on average including monochromator movement and image acquisition. Each polarization condition therefore requires about 2.8 s, yielding roughly 25 s per quadrant. Stepping between quadrants takes about 1.2 s per move, and shot-to-shot alignment takes about 6.2 s. As a result, one full shot requires about 110 s to measure. For a full wafer, which generally contains around 100-120 shots, the total measurement time amounts to approximately 3.1-3.7 hours.

4. The title of Section 3 is discussion. Actually, it is a summary of method proposed in this manuscript, which had been already provided in other sections. In the discussion section, I expect to see the discussion on the results, the limits of the proposed method, as well as the comparison of results by using different data processing, etc.

All in all, the method and results presented in the manuscript is very interesting and sound solid. however, it needs a major revision to become publishable.

We thank the reviewer for this valuable comment. In the revised manuscript, Section 3 has been substantially rewritten to function as a proper Discussion section, rather than a reiteration of the proposed methodology.

The updated discussion section now presents a detailed interpretation of the experimental results. The MM measurement of bare silicon wafer emphasize the improvement of MM spectrum consistency across the wide FOV. Building upon this foundation and the spatially dense metrology by exploiting a machine learning algorithm, we further discuss how these strengths allowed the IMMSE system to be applied to real DRAM wafer. Specifically, we highlight that the IMMSE revealed spatial variation patterns in thin film thickness, critical dimensions (CD), and overlay misalignment error that were not captured by conventional SEM. Notably, chip-level overlay statistics based on over 10 million points enabled

early detection of defective chips, demonstrating the system's potential for yield-driven process control. We suggest that these findings demonstrate the system's effectiveness in process monitoring and control, thereby addressing the core motivation and impact of the proposed approach.

In addition, we report the IMMSE's measurement accuracy in terms of the regression model's root mean square error (RMSE) for all DRAM applications presented in this study. We emphasize that this level of accuracy is sufficient for resolving nanometer-scale structural variations to control manufacturing process and current and next-generation semiconductor nodes. The IMMSE provide a 1,987-fold increase in the number of overlay measurements and approximately 662-fold faster per-point acquisition time compared to SEM (Table 4), delivering both finer spatial resolution and significantly improved throughput.

We also acknowledged the practical limitations of the IMMSE, particularly in relation to throughput. To address this, we proposed several optimization strategies such as selective wavelength sampling, reduced polarization conditions, and spatial sub-sampling. These strategies enable the IMMSE to support high-throughput process monitoring without compromising spatial resolution or measurement accuracy.

Lastly, we discuss potential applications beyond semiconductor metrology. Although the IMMSE system was originally developed for semiconductor applications, its underlying optical and analytical framework is broadly applicable to other domains. The capability of acquiring spatially dense Mueller matrix spectra over a wide field of view and broad spectral bandwidth positions the IMMSE system as a promising tool for biomedical imaging, diagnostic applications, and analytical chemistry, where high-throughput polarimetric data are also critical.

Line 388:

In this study, we experimentally demonstrated the ultra-wide-field imaging Mueller matrix spectroscopic ellipsometry (IMMSE) system for the semiconductor metrology. To achieve the wide-field metrology, we integrated a custom-designed optical system, which has a 20 mm × 20 mm field of view (FOV) with a spatial resolution of 6.5 μm. Using the IMMSE system, we measured hyperspectral image cubes, yielding over 10 million intensity spectra. For the Mueller matrix (MM) analysis, the hyperspectral image cubes were measured under multiple polarization conditions employing dual rotating polarizer configuration. By analyzing signal variations corresponding to changes in polarization states, 3×3 MM spectra from all pixels in the hyperspectral image cube were obtained.

To ensure reliable and consistent data acquisition across the wide FOV, we developed a unique algorithm to remove system-induced signal distortion. The algorithm, based on relative transmittance—referred to as system factors—compensates for signal distortions caused by spatial variations in the transmittance of optical components, such as rotating polarizers. For the clarity of the calculation, we utilized a matrix reformulation, which transforms matrix calculations of the intensity equation into scalar products.

For the experimental demonstration of the system-induced signal distortion removal, we measured the 3×3 MM spectrum of bare silicon wafer sample. After the system factors were removed, the root-mean-squared-error (RMSE) values relative to the theoretical spectrum were reduced to 2.6% of their values before removal. Similarly, the spectrum variations within the FOV were reduced to 4.7% of their values before removal.

The MM spectrum consistency across the FOV enabled spatially dense metrology for semiconductor manufacturing, providing numerous measurement points. To achieve the dense metrology, we employed a machine learning algorithm, specifically Ridge linear regression. To train the regression model, we utilized reference metrology data obtained from the conventional point-based method, such as scanning electron microscopy (SEM) and numerous MM spectra obtained using the IMMSE. The trained regression model yielded more than 10 million overlay values across the wafer, representing 12,800 times more data points compared to SEM. To evaluate the throughput of the IMMSE, we analyzed the point-by-point measurement time. The results showed that the IMMSE achieves 0.001 seconds per point, while the conventional SEM requires 0.7 seconds per point.

Leveraging its capability for the spatially dense metrology across the whole wafer area, we demonstrated that the IMMSE can be applied for the yield enhancement in semiconductor manufacturing. This was achieved by identifying spatial variations of dynamic random access memory (DRAM) structure within individual chips as well as across the wafer. The measurement results obtained using the IMMSE successfully identified distinctive spatial variation patterns in thin film thickness across the wafer. We presumed that the observed thickness

variation patterns could be interpreted as fingerprints from the manufacturing equipment. We also demonstrated that these patterns disappeared under the revised process conditions. Since the conventional point-based ellipsometry method failed to detect these abnormalities, we confidently propose that the IMMSE could serve as a unique solution for optimizing the DRAM manufacturing processes.

Furthermore, we expect that the IMMSE can be utilized to evaluate photolithography processes. To verify feasibility, we measured the overlay misalignment error of a DRAM wafer. The overlay measurement results from the IMMSE clearly identified repetitive overlay variation patterns that were undetectable in the SEM results. The results offered significant insights into the early detection of defective chips, which would be identified during the final electrical testing. Based on a numerous overlay values, we demonstrated that the IMMSE enables statistical analysis at the individual chip level, while SEM only provides fewer than two representative measurement values per chip. We showed that potential defective chips exceeding specific overlay limits can be rapidly identified from the statistical analysis.

Line 431:

In this study, we demonstrated an ultra-wide-field imaging Mueller matrix spectroscopic ellipsometry (IMMSE) system for semiconductor metrology, capable of acquiring over 10 million spatially resolved 3×3 Mueller matrix (MM) spectra across a 20 mm \times 20 mm field of view (FOV) with 6.5 μ m spatial resolution.

For the wide-FOV metrology, ensuring spectral consistency across the entire FOV area is critical. We achieved high spectrum consistency across the ultra-wide FOV, suppressing the spectrum error relative to the theoretical spectrum 94% and spectral variation across the FOV 73%. To enable this, we utilized a unique correction algorithm based on relative transmittance—referred to as system factors—which compensates for spatial non-uniformities of the signal induced from optical aberration and rotating polarizers.

Based on the high spectrum consistency across the FOV and the incorporation of a machine learning algorithm, the IMMSE enabled spatially dense metrology, revealing over 10 million metrology data across the wafer. Building upon these capabilities, practical applications of the IMMSE in DRAM manufacturing are suggested.

The experimental demonstrations were conducted on 12-inch DRAM wafers sourced from a real-world semiconductor manufacturing line, where the critical features are notably small and process variations are difficult to detect. The results highlight the IMMSE system's practical applicability and high sensitivity.

The thin film thickness, critical dimensions (CD), and overlay misalignment errors of the DRAM wafer were measured. The IMMSE enabled the detection of distinctive spatial variation patterns across the wafer, which were effectively suppressed under revised process conditions. These results demonstrate the effectiveness of the IMMSE in accurately monitoring and controlling process-induced pattern variations.

Additionally, statistical analysis of the overlay data enabled the identification of potentially defective chips exceeding the overlay thresholds. By employing the proposed technique, process abnormalities can be detected quickly and well before the electrical test, enabling rapid process feedback and contributing to yield improvement. These results suggest that the IMMSE system can serve as a powerful platform for evaluating and optimizing semiconductor manufacturing processes, ultimately contributing to yield enhancement.

The regression model employed in this study demonstrated high prediction accuracy, with root mean square error (RMSE) values below 1 nm for all evaluated parameters—including thin film thickness, CD, and overlay—as summarized in Table 3. This sub-nanometer accuracy indicates that the IMMSE system is capable of resolving nanometer-scale structural variations, which is essential for monitoring and controlling process-induced fluctuations in advanced semiconductor fabrication.

Table 3. DRAM metrology Accuracy of the IMMSE

	Film thickness		CD		Overlay	
	Process 1	Process 2	Process 1	Process 2	Process 1	Process 2
Accuracy (nm) (Reg. Model RMSE)	0.62	0.49	0.11	0.10	0.41	0.5

While the required accuracy may vary across applications, the achieved sub-nanometer precision is consistent with the metrology demands of next-generation semiconductor technologies, which continue to scale toward smaller nodes with tighter process tolerances⁵¹.

Table 4. Throughput of conventional SEM and IMMSE. The throughput is calculated based on the measurement points over full wafer area and total measurement time.

	Measurement Points (pts)	Measurement Time (s)	Throughput (s/pts)
SEM	5,994	4,196	0.700
IMMSE	11,912,908	12,600	0.001

For the overlay metrology in this study, more than 10 million overlay values were generated across a single wafer—representing a 1,987-fold increase in data points compared to scanning electron microscopy (SEM). Each point was measured in 0.001 seconds, which is significantly faster than SEM and considered acceptable for practical use in semiconductor manufacturing (Table 4).

While IMMSE provides high-density metrology data with high throughput, improving throughput is essential in practical process monitoring. To address the practical need for faster wafer-level inspection in process monitoring applications, several strategies can be considered to optimize the throughput of the IMMSE system. First, selective wavelength sampling can significantly reduce measurement time, focusing on a subset of wavelengths that are known to exhibit strong sensitivity to specific film parameters or structural variations. Second, the number of polarization conditions used for measurement can be adjusted based on the required sensitivity and complexity of the target parameter. For example, thin film thickness measurements often yield sufficient accuracy with four polarization states, and in some cases, monitoring trends over time may require only one or two states. However, it should be noted that certain applications such as overlay require 3×3 Mueller matrix reconstruction, and thus cannot be performed with reduced polarization sets. Finally, the number of measured shots across the wafer can be reduced by selecting representative regions, similar to the strategy used in conventional point-based metrology tools. This is especially effective in production environments where wafer uniformity is relatively high and full-wafer coverage is not always necessary. Together, these optimization strategies enable the IMMSE system to be tailored for high-throughput process monitoring while preserving its high-resolution and full-field measurement capabilities when needed.

While the IMMSE system was developed primarily for semiconductor metrology, its underlying optical and analytical principles make it applicable to a range of scientific and clinical domains that benefit from wide-field, high-throughput polarimetric measurements. In particular, the ability to acquire spatially dense MM data over a broad spectral range and large FOV opens possibilities in life sciences, medical diagnostics, and analytical chemistry.

In its current configuration, the IMMSE system employs linear polarization states for both generation and analysis, yielding a reduced 3×3 subset of the full MM. Despite this limitation, prior studies have shown that essential polarization parameters—such as linear diattenuation, linear retardance, and linear depolarization—can be reliably extracted from this subset^{52,53}. These parameters have been applied in biomedical contexts, including the development of flexible fiber-based 3×3 MM probes for ex vivo tissue characterization⁵⁴, and in identifying optical signatures in retinal nerve fiber layers⁵⁵, visualizing anatomical features in brain tissue⁵⁶, and highlighting characteristic patterns in cancerous tissues⁵⁷.

These applications underscore the practical utility of partial MM measurements, especially in settings requiring compact instrumentation and simplified acquisition. Looking ahead, the IMMSE system’s modular design allows for potential expansion beyond linear polarization. For instance, integrating broadband wide-field waveplates could enable full 4×4 MM acquisition, granting access to circular dichroism and comprehensive polarization characterization. Realizing this capability would require precise calibration of the wavelength- and position-dependent retardance of the waveplates, but the in-situ polarization correction framework presented in this work offers a promising foundation for addressing these challenges.

51 IEEE International Roadmap for Devices and Systems (IRDS). More Moore. IEEE Int. Roadmap for Devices and Systems, (2018). <https://irds.ieee.org>

52 Ghosh, N., Wood, M. F. G., & Vitkin, I. A. Polar decomposition of 3×3 Mueller matrix: a tool for quantitative

tissue polarimetry. *Opt. Express*, 14, 9324–9337 (2006).

53 He, H., Wang, Y., & Wang, Y. Study on the validity of 3×3 Mueller matrix decomposition. *J. Biomed. Opt.*, 20(6), 065003 (2015).

54 Alali, S., Vitkin, I. A., & Bizheva, K. Flexible polarimetric probe for 3×3 Mueller matrix measurements of biological tissue. *Sci. Rep.*, 7, 12099 (2017).

55 Bueno, J. M., & Campbell, M. C. W. Exploration of the retinal nerve fiber layer thickness by measurement of the linear dichroism. *Appl. Opt.*, 44, 7074–7081 (2005).

56 Menzel, M., Reischl, L., Axer, M., & Amunts, K. Diattenuation of brain tissue and its impact on 3D polarized light imaging. *Biomed. Opt. Express*, 8, 3163–3178 (2017).

57 Ding, H., Wang, W., Wu, Y., Boppart, S. A., & Tang, Y. Mueller matrix polarimetry for differentiating characteristic features of cancerous tissues. *J. Biomed. Opt.*, 19(7), 076013 (2014).

Reviewer #2 (Remarks to the Author):

1.The manuscript uses Ridge linear regression for the dense metrology. How does the Ridge linear regression compare with other machine learning methods, such as neural network?

We appreciate the reviewer's question. In this study, we use Ridge linear regression for metrology values, based on both practical considerations and empirical performance. Specifically, overlay-related signals were primarily extracted from off-diagonal Mueller matrix elements, such as $M_{13} + M_{31}$ and $M_{23} + M_{32}$.

Although the physical signal is relatively subtle, we observed through multiple experiments that the relationship between the spectral features and overlay is remarkably linear, especially within the typical process window. We suspect this is due to locally linear behavior in the underlying optical response, and found that Ridge regression captures this relationship effectively.

To validate this choice, we compared Ridge regression with a feedforward neural network (FFNN) composed of two hidden layers using 16 repeated train-test splits. Ridge achieved a higher mean R2 score (0.8276 vs. 0.8025) and lower RMSE (0.4509 nm vs 0.4821 nm), both statistically significant ($p < 0.01$ by Welch's t-test). These results suggest that in this specific application, Ridge not only suffices but performs better, with the added advantages of faster computation and easier interpretability.

That said, we agree that in other use cases such as critical dimension (CD) or thin film thickness (THK), where the relationship between spectral features and the target variable tends to be more nonlinear, neural networks are often more suitable and are indeed commonly used in the industry. We have added these clarifications to the revised manuscript.

Line 684:

Although other models, such as neural networks, were also considered, Ridge regression was selected due to its favorable trade-off between accuracy, computational efficiency, and interpretability. This choice was particularly suitable for the overlay measurement in Process 1 and 2 of Fig. 6, where the relationship between the MM spectra and the target parameter appeared to be locally linear.

2.Please comment on the system stability, such as thermal drift, vibration sensitivity, etc.

We appreciate the reviewer's valuable comment. Ensuring system stability is essential for utilizing IMMSE as a reliable semiconductor manufacturing equipment. Among various system components, the stability of the measured overlay is the most critical factor, as it directly impacts the control of the semiconductor manufacturing process. The stability of system factors, such as thermal drift, air flow, and polarizer error, can significantly affect the overlay stability. Therefore, we have comprehensively evaluated various aspects of system stability, including overlay results, using diverse measurement methods tailored to each system factor.

1. Temperature Stability

Temperature stability is crucial in maintaining the accuracy of our system. We measured temperature drift over 200 hours using resistance temperature detector sensors in four locations: left, top, right side of the chamber, and near the main camera. The temperature was recorded every 2 seconds. Our results show that the temperature remained under roughly $\pm 0.1^\circ\text{C}$ during the measurement period, demonstrating excellent temperature stability.

2. Image Pixel Shift

Image pixel shift can be influenced by various factors, including thermal drift, air flow, equipment vibration, and stage vibration. We examined image pixel shift using a characteristic pattern sample over 5 hours and found that it remained under ± 0.2 pixels. This pixel shift effect is contained within the noise of our Mueller matrix results.

3. Polarizer Precision

Polarizer precision is essential in maintaining the accuracy of our system. We evaluated polarizer precision by subtracting the set polarizer degree from the encoder's polarizer degree over approximately 24 hours. Our results show that all polarizer errors remained under $\pm 0.01^\circ$, demonstrating excellent polarizer precision.

4. Monochromator Stability

The stability of the monochromator is crucial for obtaining accurate spectral measurements in our system. We assessed monochromator stability, which has a wavelength error of under ± 0.24 nm from the set wavelength, with a standard deviation of about 0.001 from 250 nm to 1100 nm. These results were measured using a Bentham 487 picoammeter module over 30 repeated measurements.

5. Stage Stability

Ensuring stage stability is crucial for accurately measuring the region of interest in our system. We checked the stage XY repeatability using a Renishaw interferometer, with the stage traveling over ± 200 mm with 10 mm steps, repeated 5 times in forward and reverse directions. The results were under ± 3 μ m in 3 Sigma. Additionally, we evaluated stage Z stability, measuring it using stage Z encoder feedback during 10 seconds at 1 kHz. The result showed stability within ± 0.01 μ m.

6. The Overlay Stability

To verify our system stability, we measured the overlay between the Gate-Bit-Line and the Active region of the DRAM sample over a period of approximately 200 hours. The measurement was conducted on a single full wafer, repeated 7 times. We investigated the overlay skews between the first measured wafer and the subsequent 6 measurements. The results show that the average skew was less than 0.15 nm, and the 3σ was less than 0.56 nm across the entire wafer. The below figure illustrates the overlay and skews for both the whole wafer and specific shots. The results remained consistent throughout the 200-hour period demonstrating that the proposed IMMSE is sufficiently stable to control the sub-nm semiconductor manufacturing process over an extended period.

System Stability

Item	Thermal drift	Image pixel shift	Polarizer precision	Monochromator stability	Stage XY repeatability	Stage Z stability
Signal Range	$\pm 0.2^\circ\text{C}$	± 0.2 pixel	$\pm 0.01^\circ$	± 0.24 nm	± 3 μ m	± 0.01 μ m

The Overlay and Skew Tendency during 200 hours

Overlay Tendency in Shots during 200 hours

Skew Tendency in shots during 200 Hours

Skew Trends in shots

3. How does the measurement accuracy (or RMSE) compare with the industrial standard, e.g., the ITRS roadmap? Please comment in the applicability of the proposed method in the current and future technology nodes.

We appreciate the review's perceptive comment. As you mentioned, measurement accuracy is a crucial indicator for engineers to review results and modify the manufacturing process. Especially as device features shrink, more precise accuracy is required to serve as a basis for modifying the manufacturing process.

Regarding ITRS, it has been replaced by IRDS since 2016, and reports are published accordingly. According to the IRDS 'More Moore' section of the DRAM Technology Roadmap, the DRAM minimum half pitch is expected to scale from 13 nm in 2025 to 10 nm in 2034 [1]. Compared to the RMSE (maximum 0.62 nm) presented in this paper, our RMSE is less than 10% of the DRAM minimum half

pitch in IRDS, and this accuracy seems reasonable even in 2034. However, RMSE can vary depending on each process, the measured data (CD, structure, overlay), and the reference scale.

Although our application exhibits some variability in accuracy, it offers a significant advantage: it can pinpoint weak points across the wafer that are caused by manufacturing processes. These weak points cannot be detected using traditional low-frequency measurement data. Our application's wide coverage and dense data points for critical dimension (CD), structure, and overlay measurements enable it to provide a more comprehensive understanding of wafer characteristics. The unique strengths of IMMSE lie in its ability to extract valuable insights from massive amounts of data, making it an indispensable metrology tool in semiconductor manufacturing, not only for current but also for future process nodes.

We have added further discussion on measurement accuracy in the Discussion section as follows.

Line 454:

The regression model employed in this study demonstrated high prediction accuracy, with root mean square error (RMSE) values below 1 nm for all evaluated parameters—including thin film thickness, critical dimensions (CD), and overlay—as summarized in Table 3. This sub-nanometer accuracy indicates that the IMMSE system is capable of resolving nanometer-scale structural variations, which is essential for monitoring and controlling process-induced fluctuations in advanced semiconductor fabrication.

While the required accuracy may vary across applications, the achieved sub-nanometer precision is consistent with the metrology demands of next-generation semiconductor technologies, which continue to scale toward smaller nodes with tighter process tolerances¹.

Table 3. DRAM metrology Accuracy of the IMMSE

	Film thickness		CD		Overlay	
	Process 1	Process 2	Process 1	Process 2	Process 1	Process 2
Accuracy (nm) (Reg. Model RMSE)	0.62	0.49	0.11	0.10	0.41	0.5

[1] IEEE International Roadmap for Devices and Systems, "More Moore." Institute of Electrical and Electronics Engineers, 2021, doi: 10.1109/IRDS54852.2021.00010.

4. In Table 3, the authors compared the throughput of their IMMSE system with SEM. IMMSE took 3.5 hours to complete the measurement. Although IMMSE measurement contains much more data point, in reality, it often is more important to complete the wafer inspection faster for overall throughput particularly when the measurement is used for process monitoring. Can the authors comment on how to further improve the total measurement time? For example, can one reduce the measurement point to get much shorter measurement time?

We appreciate the reviewer's insightful comment. We agree that while IMMSE provides high-density spectral and polarization data, improving throughput is essential for practical process monitoring scenarios. Below we summarize several strategies that can significantly reduce the total measurement time, especially when the goal is not full initial characterization but rather ongoing monitoring of process stability or yield.

1. Spectral range optimization

For process monitoring, it is often possible to identify specific wavelengths that show strong correlation with the physical parameter of interest. (e.g., film thickness or CD variation). Because the time spent acquiring spectral images dominates over motor movement time, reducing the number of wavelength points can almost linearly reduce the total measurement time. Selective wavelength sampling has proven effective in our internal monitoring use cases.

2. Reduction of polarization conditions

Full Mueller matrix acquisition may not be necessary in all monitoring scenarios. For example, in thin film thickness measurement, accurate results can often be obtained using as few as 4 polarization states, rather than the default 9. In some cases, two or even a single polarization condition may be sufficient for trend tracking, especially when the objective is anomaly detection or process drift monitoring. However, we note that certain advanced metrics – such as overlay– require full polarization information and may not be reliably measured with reduced polarization states. Therefore, the number of polarization conditions should be selected based on the specific monitoring goal and required sensitivity

3. Shot-level subsampling

When full-wafer coverage is not required, the measurement can be limited to a subset of representative shots, similar to conventional point-based metrology tools. This is particularly applicable when the wafer shows uniformity across regions and the goal is to catch outliers or verify process stability.

4. Hardware-level improvements

Additional speed gains can be expected from future hardware upgrades. For example, using a stronger light source can improve SNR, allowing for shorter exposure time or reduced spectral sampling. Currently, we use an EQ77 light source; a higher-power model (EQ850) is under development and could enhance throughput. Further improvements can be achieved through better maintenance of fiber coupling efficiency, and by upgrading motor systems for both the PSG/PSA rotation and monochromator grating positioning.

These strategies are being actively explored and tailored for different use cases. We have added a summary of these possibilities to the revised Discussion section.

Line 466:

While IMMSE provides high-density metrology data with high throughput, improving throughput is essential in practical process monitoring. To address the practical need for faster wafer-level inspection in process monitoring applications, several strategies can be considered to optimize the throughput of the IMMSE system. First, selective wavelength sampling can significantly reduce measurement time, focusing on a subset of wavelengths that are known to exhibit strong sensitivity to specific film parameters or structural variations. Second, the number of polarization conditions used for measurement can be adjusted based on the required sensitivity and complexity of the target parameter. For example, thin film thickness measurements often yield sufficient accuracy with four polarization states, and in some cases, monitoring trends over time may require only one or two states. However, it should be noted that certain applications such as overlay require 3×3 Mueller matrix reconstruction, and thus cannot be performed with reduced polarization sets. Finally, the number of measured shots across the wafer can be reduced by selecting representative regions, similar to the strategy used in conventional point-based metrology tools. This is especially effective in production environments where wafer uniformity is relatively high and full-wafer coverage is not always necessary. Together, these optimization strategies enable the IMMSE system to be tailored for high-throughput process monitoring while preserving its high-resolution and full-field measurement capabilities when needed.

5. It might be better if a flowchart of data processing and the overall algorithm schematic can be included to clarify these various aspects.

We appreciate the reviewer's helpful suggestion to include a flowchart to clarify the data processing pipeline and overall algorithmic structure of the IMMSE system. In response, we have updated Fig. 1c to enhance the clarity of the full data processing workflow in this study—from data acquisition, through Mueller matrix calculation, to achieving spatially dense metrology based on a machine learning algorithm.

The corresponding description in the manuscript has also been revised to align with the updated schematic and to improve the overall readability of the manuscript.

(Original Figure)

Fig. 1. Configuration of the IMMSE system and data acquisition process. **a** A schematic of the IMMSE system, which consists of light source, monochromator, illumination optics, imaging optics, image sensor, polarization manipulation module (indicated as PSG and PSA) and align optics. **b** A realized optical setup implemented in a metrology equipment. We developed the metrology equipment integrated with the optical setup, a wafer transport robot and a stage system for wafer movement. **c**. The data acquisition sequence using the IMMSE. Ultra-wide-field hyperspectral images with a 20 mm × 20 mm FOV under multiple polarization conditions were measured using the monochromator and the polarization manipulation module. By analyzing signal variations corresponding to the polarization state changes, the 3×3 MM spectrum can be derived.

(Revised Figure)

Fig. 1. Configuration of the IMMSE system and its process flow. **a** Schematic of the IMMSE system, which consists of light source, monochromator, illumination optics, imaging optics, image sensor, polarization manipulation module (indicated as PSG and PSA) and align optics. **b** Photograph of the realized optical setup implemented within the metrology tool, which also incorporates a wafer transport robot and a stage system for wafer handling. **c** Data acquisition and processing flow of the IMMSE system. Hyperspectral image cubes under multiple polarization conditions were acquired. The system-induced signal distortion, referred to as the system factor, was corrected in the acquired signal. The 3×3 Mueller matrix (MM) spectrum was derived by analyzing signal variations under different polarization states. The spatially dense metrology was achieved by utilizing a regression model trained on reference metrology data from conventional SEM and the acquired spatially resolved MM spectra.

Line 99:

Fig. 1c shows the data acquisition sequence of the proposed system. In this study, we acquired a hyperspectral image cube with an ultra-wide field of view (FOV) of $20 \text{ mm} \times 20 \text{ mm}$ and 35 wavelengths, covering from 270 nm to 780 nm at 15 nm intervals. Since the obtained hyperspectral image cube has 3200×3200 pixels with a pixel size of $6.5 \text{ }\mu\text{m}$, over 10 million intensity spectra can be measured in the hyperspectral image cube. Utilizing the polarization manipulation module, hyperspectral image cubes were measured under multiple polarization conditions. By analyzing the signal variations corresponding to the polarization changes, the MM spectrum can be calculated for all pixels in the image.

Line 107:

Fig. 1c shows the data acquisition and processing flow of the proposed IMMSE system. In this study, we acquired a hyperspectral image cube with an ultra-wide field of view (FOV) of $20 \text{ mm} \times 20 \text{ mm}$ under multiple wavelength conditions. Given its high spatial resolution of 3200×3200 pixels and a pixel size of $6.5 \text{ }\mu\text{m}$, the hyperspectral cube contains over 10 million individual intensity spectra in a single acquisition. Utilizing the polarization manipulation module, hyperspectral image cubes were measured under multiple polarization conditions. The obtained signals were calibrated using unique data processing algorithm to correct for signal distortion induced by the measurement system, including rotating polarizer and aberrations including image distortion. The MM spectrum can be calculated for all pixels in the calibrated image by analyzing the signal variations corresponding to the polarization changes. Using the spatially resolved Mueller matrix (MM) spectra, a regression model was trained with reference data from conventional metrology tools such as SEM. This approach enabled high-density, wafer-scale metrology of nanostructures.

Reviewer #3 (Remarks to the Author):

This paper proposes an ultra-wide imaging Mueller matrix ellipsometer for semiconductor metrology. By achieving a 20mm×20mm FOV with a spatial resolution of 6.5μm, they can obtain more than 10 million 3×3 Mueller matrix spectra to implement spatially dense semiconductor metrology. The authors have demonstrated impressive results in identifying spatial variations of the DRAM overlay to potentially improve the semiconductor manufacturing yield.

However, after carefully reviewing this manuscript, in my opinion,

this paper is only an incremental innovation in technique, and lacks original novelty in theory and method.

The imaging Mueller matrix ellipsometry is a wide-spread and well-published technique, and its applications in semiconductor metrology for thin films, nanostructures, even overlays are also well published.

Machine learning assisted data analysis for ellipsometry especially for complex multilayer thin films and nanostructures has also been well studied during recent years.

The authors should clarify and strengthen their original novelty compared with published techniques. I cannot recommend the current manuscript for the publications in Nature Communication.

This paper proposes some detailed techniques for their configuration of imaging MMSE, such as the calibration of system-induced signal distortion, and the residual polarization in light source due to the rotating polarizer in the PSG and polarization sensitivity of the detector due to the rotating polarizer in the PSA by using so-called system factor analysis. I think it can be published in a specific journal in fields of optics or instruments. But more details about their configurations and techniques for how to achieve such an ultra-wide FOV with a relatively decent resolution should be provided.

We thank the reviewer for pointing out the importance of clearly stating the novelty of our work. As the reviewer correctly noted, imaging based Mueller matrix (MM) ellipsometry and its applications to nanostructures are already well established in the literature. We have identified several reference studies that have explored related topics. That said, we believe our systematic analysis of spectral variation across a wide field of view offers a unique perspective that has not been fully addressed in previous works.

The key contribution of our work lies in the realization of a MM measurement system that ensures high signal consistency across a large-area field of view (FOV, 20 mm × 20 mm). This achievement enables the extension of MM ellipsometry techniques to large-area, high-throughput applications in the semiconductor metrology.

The proposed method is based on the concept of relative transmittance. Since many forms of signal distortion appear as intensity variations, this approach systematically accounts for a range of distortion sources—including polarization state, phase retardation, and optical aberrations.

Building on this framework, our approach applies signal optimization globally—across all optical conditions, wavelengths, polarization states, and every pixel within the wide FOV—ensuring robust and reliable MM measurements over large measurement areas.

In Section 2.4 (Mueller matrix of bare silicon wafer), the improvement in spectral consistency across the ultra-wide FOV is clearly demonstrated, showing a 94% reduction in spectrum error relative to the theoretical spectrum and a 73% reduction in spectral variation across the FOV.

By achieving a high level of MM spectrum consistency across the ultra-wide FOV, the IMMSE system enables machine learning–based spatially dense metrology with high spatial resolution, sub-nanometer accuracy, and fast measurement throughput across the semiconductor wafer.

The regression model employed in this study demonstrated sub-nanometer accuracy, with root mean square error (RMSE) values below 1 nm for all evaluated results—including thin film thickness, critical dimensions (CD), and overlay—as summarized in the Table 3.

Table 3. DRAM metrology Accuracy of the IMMSE. The metrology accuracy was evaluated using the RMSE error of the regression model.

	Film thickness		CD		Overlay	
	Process 1	Process 2	Process 1	Process 2	Process 1	Process 2
Accuracy (nm) (Reg. Model RMSE)	0.62	0.49	0.11	0.10	0.41	0.5

In the case of overlay measurements, the IMMSE system produced over 10 million data points across a single wafer—representing a 1,987-fold increase compared to scanning electron microscopy (SEM)—while achieving a per-point measurement time of 0.001 seconds, more than 662 times faster than SEM. This performance enables full-wafer metrology to be completed within 3.5 hours, as summarized in the Table 4.

Table 4. Throughput of conventional SEM and IMMSE. The throughput is calculated based on the measurement points over full wafer area and total measurement time.

	Measurement Points (pts)	Measurement Time (s)	Throughput (s/pts)
SEM	5,994	4,196	0.700
IMMSE	11,912,908	12,600	0.001

In particular, the sub-nanometer accuracy and extremely high throughput enable precise quantification of critical parameters including thin film thickness, CD, and overlay misalignment error of the DRAM wafer. Notably, our study does not rely on wafers with simple patterns commonly used in laboratory settings; instead, we performed measurements directly on 12-inch DRAM wafers that were actively being produced in a high-volume semiconductor fabrication line. This challenging and industrially relevant application highlights the practical novelty and robustness of our approach.

These wafers contain structures with feature sizes ranging from several to tens of nanometers, and their process-induced variations often fall within the sub-nanometer range—posing a significant challenge for conventional imaging-based ellipsometry. Despite this difficulty, our approach successfully captured these extremely small variations with high accuracy and full-wafer coverage, all within a measurement time that is acceptable from the perspective of manufacturing throughput.

This capability enables detection of spatial variation patterns that are impossible to observe with conventional point-based metrology tools such as SEM, thereby supporting advanced process monitoring and control. We believe that the combination of full-wafer coverage, sub-nanometer precision, and demonstrated applicability to real semiconductor production with high throughput represents the core novelty of our work.

We also implemented additional strategies to achieve wide-FOV metrology. On the hardware side, we adopted an Offner-type reflective imaging optical system specifically designed to minimize aberrations across the ultra-wide FOV, and employed a low-NA illumination system to ensure uniform illumination while minimizing spectral sensitivity degradation.

On the algorithm side, we applied camera linearity correction to ensure accurate intensity response across the wide imaging area, and distortion correction algorithms to accurately map spatial positions and reduce geometric artifacts in the acquired images.

These combined innovations establish a highly reliable data acquisition framework, which not only supports high-accuracy, wide-area metrology using IMMSE, but also enables practical deployment in semiconductor production.

Lastly, we compared previously published imaging-based ellipsometry techniques for nanoscale structural measurements, as summarized in the following tables. Compared to the referenced methods, the IMMSE offers superior performance in terms of FOV and spectral range. Moreover, the approach

employed to ensure high reliability across such a broad FOV and wavelength range is more broadly applicable and practical.

Previous studies have attempted to optimize specific optical imperfections—such as phase retardation of rotating compensators [1], initial angular offset of polarizers and compensators across wavelengths [2], or nulling-based optimization [3]. In contrast, the IMMSE adopts a more generalizable strategy that enables comprehensive control over signal distortions from illumination path, imaging optics, and polarizers. It enables consistent and reliable measurements across a wide FOV and broad spectral range. This capability is critical for accurately characterizing fine structural variations across DRAM wafers, whereas prior studies have largely been limited to standardized samples or contrast-based image demonstrations [1–5].

[1] Lianhua Jin, Yuki Iizuka, Takashi Iwao, Eiichi Kondoh, Makoto Uehara, and Bernard Gelloz, "Calibration of the retardation inhomogeneity for the compensator-rotating imaging ellipsometer," *Appl. Opt.* 58, 9224-9229 (2019)

[2] Chen, Xiuguo, et al. "Development of a spectroscopic Mueller matrix imaging ellipsometer for nanostructure metrology." *Review of Scientific Instruments* 87.5 (2016).

[3] Braeuninger-Weimer, Philipp, et al. "Fast, noncontact, wafer-scale, atomic layer resolved imaging of two-dimensional materials by ellipsometric contrast micrography." *ACS nano* 12.8 (2018): 8555-8563.

[4] Jin, L., et al. "Rotatable Offner imaging system for ellipsometric measurement." *Review of Scientific Instruments* 88.1 (2017).

[5] Sigger, Florian, et al. "Spectroscopic imaging ellipsometry of two-dimensional TMDC heterostructures." *Applied Physics Letters* 121.7 (2022).

Table 3. Comparison of representative imaging Mueller matrix ellipsometry systems. This table compares key features of the proposed system and three representative imaging Mueller matrix ellipsometry systems reported in the literature.

	This work	Jin et al. [1]	Chen et al. [2]	Braeuninger-Weimer et al. [3]
System Configuration	Dual rotating polarizer (3×3 MM)	PCrSA (Δ, ψ)	Dual rotating compensator (4×4 MM)	Fixed PCSA (Film model parameters)
Demonstrated Wavelength (nm)	250-1100	650	400-700	390, 450, 550
Field of View (mm ²)	20×20	5.7×4.3	4×4	2×2
Residual Spectral Error	0.004	-	0.01	-
Inhomogeneity Correction Items	Polarizer transmittance Polarization sensitivity of detection path Polarization state of illumination on PSA	Retardation	- (Pixel averaged system parameters)	-

In overall consideration of the reviewer's comments, we recognized that the manuscript required substantial revision to more clearly demonstrate the novelty of the IMMSE system. The following changes have been made in the revised manuscript, along with the purpose of each revision:

1) To emphasize the novelty of the IMMSE, we revised the results presented in Section 2.5 (DRAM Application). The revised manuscript more clearly demonstrates that the IMMSE can measure DRAM wafers processed in a real fabrication line, with detailed descriptions of the process parameters. It also explains why such measurements are both important and technically challenging, due to the extremely small feature sizes and subtle process-induced variations across the wafer.

(Original Figure)

Fig. 5. Thin layer thickness metrology of DRAM wafer and analysis. **a** The thickness measurement results of a wafer processed under the specific CVD process (designated as Process 1) using the conventional point-based ellipsometry technique. It only reveals the limited thickness trends. **b** IMMSE measurement results of the same wafer sample, shown in (a). The result shows distinct thickness variation patterns over the whole wafer area, presumed to be caused by the manufacturing equipment. **c** IMMSE measurement results of the wafer sample processed under revised process conditions (referred to as Process 2).

(Revise Figure)

Fig. 4. Thin layer thickness metrology of a DRAM wafer and comparative analysis. **a** Schematic diagram of thin-film deposition using a gas chamber equipped with multiple nozzles arranged in a hexagonal array. **b** Thickness measurement results of a wafer processed under the initial CVD condition (designated as Process 1), obtained using a conventional point-based ellipsometry technique. This method reveals only limited information about the overall thickness distribution. **c** IMMSE-based thickness measurement of the same wafer, showing distinct non-uniformity patterns across the full wafer area, which are presumed to originate from the characteristics of the nozzles array. **d** IMMSE measurement results for the wafer processed under revised process conditions (Process 2), showing improved film thickness uniformity across the wafer.

Line 274:

The chemical vapor deposition (CVD) is one of the important manufacturing processes in semiconductor fabrication, forming thin layers on the surface of semiconductor wafers^{48,49}. Fig. 5a illustrates the thickness measurement results of a wafer processed under the specific CVD process conditions (indicated as Process 1). The measurements were taken at 49 points across the wafer using a conventional point-based ellipsometry system, revealing only a global thickness trend due to its sparse sampling.

In contrast, the thickness measurement of the same sample wafer using the IMMSE reveals distinct, repetitive patterns throughout the wafer, as shown in Fig. 5b. The spatially dense metrology, with more than 7 million points, enables the identification of unique thickness variations that the conventional point-based method fails to detect. These patterns could be interpreted as fingerprints of the manufacturing equipment and offers invaluable insights for the process optimization to enhance the yield.

Fig. 5c illustrates the measurement results of a wafer processed under modified CVD conditions (referred to as Process 2). Unlike Fig. 5b, Fig. 5c exhibits no distinctive patterns, implying that the modified process conditions are better suited for this manufacturing step. Based on the experimental results, we propose that the IMMSE can be utilized to evaluate and optimize manufacturing processes, as it offers significant insights into the process conditions.

Line 290:

Chemical vapor deposition (CVD) is one of the essential manufacturing processes in semiconductor fabrication, used to form thin layers on the surface of semiconductor wafers^{47,48}. As illustrated in Fig. 4a, the schematic of the CVD setup shows thin-film deposition inside a gas chamber equipped with multiple nozzles arranged in a hexagonal array, which affects the spatial uniformity during deposition.

Figure 4b presents the thickness measurement results of a wafer processed under a specific CVD condition (indicated as Process 1). The measurements were acquired at 49 points across the wafer using a conventional point-based ellipsometry system, which reveals only coarse global thickness trends due to its sparse spatial sampling.

In contrast, the thickness measurement of the same sample wafer using the IMMSE reveals distinct, repetitive patterns throughout the wafer, as shown in Fig. 4c. The spatially dense metrology, with more than 7 million points, enables the identification of unique thickness variations that the conventional point-based method fails to detect. These patterns could be interpreted as fingerprints of the manufacturing equipment and offer invaluable insights for the process optimization to enhance the yield.

Fig. 4d illustrates the measurement results of a wafer processed under modified CVD conditions (referred to as Process 2). Unlike Fig. 4c Fig. 4d exhibits no distinctive patterns, implying that the modified process conditions are better suited for this manufacturing step.

(Original Figure)

Fig. 6. Overlay metrology of DRAM wafer and analysis a,b Overlay measurement using the conventional SEM (a) over the full wafer area and (b) over four adjacent shot areas marked with a dashed box in (a). c,d IMMSE measurement results of the same sample wafer, shown in (a) and (b). (c) shows the same measurement area as (b), which exhibits overlay variation patterns (indicated by black arrows) repeated in each shot. e,f IMMSE overlay measurement results of another sample wafer processed under modified conditions. g The histogram of overlay values within a single chip area marked by a red box in (c) and (f). h Distribution of the mean and standard deviation of overlay values for 100 chips within the area depicted in (c) and (f).

(Revised Figure)

Fig. 6. Overlay metrology of a DRAM transistor structure and comparative analysis between conventional SEM and IMMSE a 3D schematic of the DRAM transistor structure. b SEM image showing the BL; horizontal lines and CM; circular patterns beneath the BL from a top-down view, corresponding to the black arrow and shaded plane in (a). c TEM image of the same structure viewed from the side, as indicated by the red arrow and shaded plane in (a). The overlay error is evaluated by measuring the vertical misalignment between the BL and CM layers. d Overlay measurements acquired using a conventional SEM across the wafer processed under Process 1. e Enlarged view of four adjacent shot regions indicated by the dashed box in (d), showing the local overlay distribution in more detail. f IMMSE overlay measurement results of the same region shown in (e), revealing recurring overlay variation patterns within each shot, as marked by black arrows. g Full-wafer IMMSE measurement results corresponding to the area shown in (d), providing high-density overlay information for Process 1. h, i IMMSE overlay measurements of a different wafer processed under modified conditions (Process 2), demonstrating improved overlay performance and reduced variation. j Histogram of overlay values within a single chip area, marked by red boxes in (f) and (i), comparing Process 1 and Process 2 results. k Distribution of the mean and standard deviation of overlay values for 100 chips located within the region indicated in (f) and (i), showing statistical improvement in overlay control under Process 2.

Line 324:

Photolithography is one of the key technologies in semiconductor manufacturing. However, with the increasing complexity of device structures, achieving precise patterning has become increasingly challenging⁵⁰. By utilizing the IMMSE technique, we suggest that the system could serve as a unique solution for optimizing photolithography processes.

Fig. 6a and 6b represents the SEM measurement results of the misalignment between the GBL and the DCC after GBL etching of a DRAM wafer processed under specific process conditions (referred to as Process 1). Approximately 6,000 points across the wafer and 261 points within four adjacent shot areas were measured. Notably, as shown in Fig. 6c and 6d, the results measured by the IMMSE reveal repetitive overlay variation patterns on the left side of each shot (indicated by black arrows in Fig. 6c), whereas the SEM result fail to capture these local variations. As shown in Fig. 6e and 6f, the IMMSE measurement results for another sample wafer processed under revised lithography conditions (referred to as Process 2) indicate the absence of overlay variation patterns. This implies that Process 2 is a more mature process condition compared to Process 1.

The overlay variations observed in Fig. 6c could potentially be detected during electrical testing after the entire manufacturing process is completed, where chips located in the same position within each shot are consistently identified as functional or defective. By employing the proposed technique, process abnormalities can be

detected quickly and well before the electrical test, enabling rapid process feedback and contributing to yield improvement.

In addition to these qualitative analysis of the manufacturing process based on the overlay maps, the IMMSE enables quantitative analysis by leveraging a large amount of measurement points, even at the individual chip level. Fig. 6g presents a histogram of the overlay distribution within the single chip area outlined by a red box in Fig. 6c and 6f. The result under the Process 1 exhibits an overlay mean offset of -1.31 nm with a standard deviation of 0.36 nm, whereas the Process 2 shows an overlay mean value close to zero (-0.03 nm) and a standard deviation of 0.24 nm. The obtained statistical analysis can be applied to any localized region on the wafer. Fig. 6h illustrates the distribution of the mean and standard deviation values of overlay for cell blocks within each of 100 chips in the region depicted in Fig. 6c and 6f. The result indicates that the distribution of mean and standard deviation values of the Process 2 are significantly reduced compared to those of the Process 1. Based on the statistical analysis, potentially defective chips exceeding specific overlay limits can be intuitively identified.

Line 351:

Fig. 6a illustrates the DRAM transistor structure and the principle of overlay evaluation. The overlay error measured in this study refers to the misalignment between the bit line (BL) and the contact material (CM) to the drain of the transistor, which are vertically stacked structures. The BL delivers read/write signals, while the CM connects the drain region of the cell transistor to the BL. Misalignment between these two layers can lead to incomplete or unstable electrical contact, potentially resulting in functional failure or increased resistance. Therefore, precise overlay control is critical to ensure reliable device operation and high manufacturing yield⁵⁰.

Fig. 6b shows a top-view SEM image corresponding to the black arrow and shaded plane in Fig. 6a. The overlay between the BL and CM is estimated by measuring the relative distance between the two features. Fig. 6c presents a side-view TEM image that reveals extremely narrow gaps between adjacent structures. The combination of ultra-small feature sizes and sub-nanometer misalignment, as illustrated in Fig. 6a–c, poses a significant challenge for overlay metrology, especially in achieving both high sensitivity and wafer-scale coverage.

Fig. 6d presents the overlay measurement results obtained using a conventional SEM for a wafer processed under specific lithography conditions (referred to as Process 1), with approximately 6,000 points measured across the wafer. Fig. 6e shows an enlarged view of the same SEM data, focusing on four locally adjacent shot regions comprising 261 measurement points.

Fig. 6f and 6g show the IMMSE overlay measurements of the same wafer. While the SEM results provide only 6,000 points across the wafer and 261 points within the four localized shot areas, the IMMSE yields more than 10 million measurements for the entire wafer and over 100,000 measurements within the same localized region.

Leveraging its massive number of overlay measurements, distinct variation patterns on the left side of each shot area can be clearly identified, as indicated by the black arrows in Fig. 6f. These localized variations remain undetected in the SEM results but are readily observed through the dense spatial sampling enabled by IMMSE.

To evaluate the effect of process optimization capability of the IMMSE, a second wafer processed under revised lithography conditions (referred to as Process 2 in the Fig.6) was analyzed. The IMMSE results for this wafer, shown in Fig. 6h and 6i, indicate that the shot-level overlay variation pattern is no longer present, suggesting improved lithography performance and greater process maturity compared to Process 1.

In addition to these qualitative analyses of the manufacturing process based on the spatially dense overlay maps, the IMMSE also enables quantitative evaluation of overlay variation within individual chips by utilizing a large number of measurement points. Fig. 6j presents a histogram of the overlay distribution within a single chip area outlined in red box in Fig. 6f and 6i. The result under Process 1 exhibits a mean offset of -1.31 nm with a standard deviation of 0.36 nm, whereas Process 2 yields a near-zero mean value of -0.03 nm and a reduced standard deviation of 0.24 nm.

The obtained statistical analysis can be applied to any localized region on the wafer. Fig. 6k illustrates the distribution of the mean and standard deviation of overlay values computed from cell blocks within each of 100 chips located in the regions shown in Fig. 6f and 6i. The result indicates that the distribution of both metrics in Process 2 is significantly improved compared to those in Process 1.

2) We have newly added CD measurement results to emphasize that the IMMSE can be applied to complex 3D structures in DRAM devices. The added results present the CD measurements of the DRAM capacitor structure, including detailed process information and demonstrating the potential of IMMSE for process monitoring by capturing spatial CD variation patterns across the wafer at the sub-nanometer scale. Details of the regression model used for the CD measurements have also been added in Section 4.5 (Machine learning for spatially dense metrology) of the revised manuscript.

(Newly Added Figure)

Fig. 5. CD metrology of a DRAM wafer and analysis. **a** Schematic illustration of the DRAM structure. **b** SEM image corresponding to the cross-sectional plane illustrated as black dashed box in **a**. **c** CD measurement result obtained using a conventional SEM. The wafer was processed under the regular etch process referred to as Process 1 in the figure. **d** CD measurement result acquired using the IMMSE. While the SEM provides only 536 measurement points, the IMMSE yields over 10 million CD data points across the entire wafer. This dense spatial sampling reveals distinct circular CD variation patterns (black dashed circles) that are not captured in the SEM results. **e** CD measurement result of the wafer processed under the revised etch process referred to as Process 2 using a conventional SEM. **f** CD measurement result of the same wafer acquired using the IMMSE. The circular CD variation pattern observed in Process 1 has been eliminated, and the overall CD uniformity across the wafer is improved under Process 2.

Line 306:

The DRAM capacitor structure features a high aspect ratio trench filled with metallic materials (Fig. 5a), which is critical for achieving the desired electrical features within a limited cell footprint. Due to this geometry, precise control of the etch process before the metal fill is essential to ensure uniform depth and profile across the wafer. In particular, the CD measured at the top surface after etching serves as an indirect but effective indicator for assessing the quality and uniformity of the deep etch process⁴⁹ (Fig. 5b).

Fig. 5 presents a comparative analysis of CD measurements obtained using the conventional SEM and the proposed IMMSE system for DRAM wafers processed under two different etch conditions. For the wafer processed under the regular etch process referred to as Process 1, the SEM-based CD measurements (Fig. 5c) are limited to 536 points, which is insufficient to capture global CD variation trends. In contrast, the IMMSE-based measurements (Fig. 5d) provide over 10 million CD values across the full wafer, revealing concentric CD variation patterns (black dashed circles in Fig. 5d) that remain hidden in the SEM results due to sparse sampling. It is noteworthy that the proposed method can identify CD variation patterns at the sub-nanometer scale.

A revised etch condition (Process 2) was applied to suppress these spatial variations. As seen in Fig. 5e, the SEM result does not adequately reflect changes in global CD distribution. However, the IMMSE result (Fig. 5f) confirms that the circular variation pattern has been effectively eliminated, and the overall CD uniformity is significantly improved across the wafer. These results demonstrate the capability of the IMMSE system to enable dense, wafer-scale metrology, offering critical insights into etch process performance that are not accessible through conventional SEM-based sampling.

Line 661:

In Section 2.5, the reference overlay data consisted of 815 points, with 20% allocated to the test set. The remaining data was used to determine an appropriate regularization constant through 4-fold cross-validation. A total of 280 features, derived from 35 wavelengths and 8 MM components, were utilized to train the

regression model. The optimized regularization constant was determined to be 0.0012. The performance of the regression model was evaluated using the test set which achieved an R-squared (R2) value of 0.93 and a root-mean-square-error (RMSE) of 0.62 nm.

For the thin layer measurement described in Section 2.6, the reference dataset was limited to only 49 points. To address this, we repeated the data splitting process 1,000 times, keeping the test set size at 30% to determine the optimal data split configuration. We exclusively used the m_{33}/m_{22} spectrum, corresponding to the C-spectrum in conventional ellipsometry⁴⁵, due to the isotropic nature of the thin film layer, where the off-diagonal MM components are expected to be zero. Despite using the single MM spectrum, the dataset size remains small, consisting of 35 wavelengths and 17 training samples. To address this limitation, we employed principal component analysis (PCA) and reduced the dimensionality of the dataset's features from 35 to 16. The optimized regularization constant of the regression model for the wafer sample processed under the Process 1 was determined to be 0.00018, yielding an R2 score of 0.21 and an RMSE of 0.62 nm on the test set. Although this result showed a low R2 value, it was sufficient for identifying the unique thickness variation trends across the entire wafer area. For the wafer sample processed under the Process 2, the regression model was optimized with a regularization constant of 0.000035, achieving an R2 score of 0.95, and an RMSE of 0.49 nm on the test set.

The data splitting for the overlay measurement described in Section 2.6 was performed under the same conditions described in Section 2.5. For the regression model trained on the wafer processed under the Process 1, the optimized regularization constant was 0.00006, with an R2 score of 0.74 and an RMSE of 0.41 nm on the test set. The regression model for the sample processed through the Process 2 was optimized with a regularization constant of 0.0009, yielding an R2 score of 0.50 and an RMSE of 0.36 nm on the test set.

Line 637:

For the thin layer measurement described in Section 2.5, the reference dataset was limited to only 49 points. To address this, we repeated the data splitting process 1,000 times, keeping the test set size at 30% to determine the optimal data split configuration. We exclusively used the m_{33}/m_{22} spectrum, corresponding to the C-spectrum in conventional ellipsometry⁴⁵, due to the isotropic nature of the thin film layer, where the off-diagonal MM components are expected to be zero. Despite using the single MM spectrum, the dataset size remains small, consisting of 35 wavelengths and 17 training samples. To address this limitation, we employed principal component analysis (PCA) and reduced the dimensionality of the dataset's features from 35 to 16. The optimized regularization constant of the regression model for the wafer sample processed under the Process 1 was determined to be 0.00018, yielding an R2 score of 0.21 and an RMSE of 0.62 nm on the test set. Although this result showed a low R2 value, it was sufficient for identifying the unique thickness variation trends across the entire wafer area. For the wafer sample processed under the Process 2, the regression model was optimized with a regularization constant of 0.000035, achieving an R2 score of 0.95, and an RMSE of 0.49 nm on the test set.

In the measurement of the DRAM capacitor described in Section 2.5, the reference CD dataset consisted of 536 points, with 20% allocated to the test set. The remaining data was used to determine an appropriate regularization constant through 4-fold cross-validation. A total of 280 features, derived from 35 wavelengths and 8 MM components, were utilized to train the regression model. For the regression model trained on the wafer processed under the Process 1, the optimized regularization constant was 0.015, with an R2 score of 0.715 and an RMSE of 0.106 nm on the test set. The regression model for the sample processed through the Process 2 was optimized with a regularization constant of 0.021, yielding an R2 score of 0.589 and an RMSE of 0.103 nm on the test set.

The data splitting for the overlay measurement in the same section was performed under the same conditions with the CD measurement. For the regression model trained on the wafer processed under the Process 1, the optimized regularization constant was 0.00006, with an R2 score of 0.74 and an RMSE of 0.41 nm on the test set. The regression model for the sample processed through the Process 2 was optimized with a regularization constant of 0.0009, yielding an R2 score of 0.50 and an RMSE of 0.36 nm on the test set.

3) In the original version of the manuscript, Section 2.5 (Machine Learning–aided spatially dense metrology) presented overlay measurement results. In the revised manuscript, however, Section 2.6 (DRAM metrology and applications) has been updated to include detailed descriptions of all measured process parameters, including thin film thickness, CD, and overlay misalignment. To improve the logical flow and avoid redundancy, we removed the original Section 2.5 and changed 2.6 (DRAM metrology and applications) to Section 2.5.

On the other hands, we revised the beginning of Section 2.5 (DRAM Applications) to better explain the principle and implementation of spatially dense metrology, in order to address the potential discontinuity caused by the removal of the previous section. The removed content has now been relocated and incorporated into Section 4.4 (Data acquisition procedure)

Line 257:

2.6. DRAM metrology and applications

We demonstrated that the proposed IMMSE technique provides reliable MM spectra across the entire wafer area and delivers a numerous metrology values with exceptionally high spatial density and throughput. These advantages enable a detailed analysis of device structure variations within individual chips as well as across the entire wafer. In this section, we propose practical applications of the IMMSE for the DRAM manufacturing process.

Line 264:

2.5. DRAM metrology and applications

We demonstrated that the proposed IMMSE technique provides reliable and consistent MM spectra over an ultra-wide FOV. To enable spatially dense metrology based on these spectra, we employed a machine learning algorithm—specifically, Ridge regression—which was trained using both the acquired massive number of MM spectra and reference data obtained from conventional point-based metrology tools such as SEM. This approach allowed us to achieve wafer-scale characterization of nanostructures, with significantly higher spatial sampling density compared to traditional methods (See section 4.4 and 4.5 for the detailed information).

In this section, we present practical applications of the IMMSE in DRAM manufacturing, demonstrating its effectiveness in analyzing structural variations and detecting process-induced deviations across the wafer with high spatial resolution.

Line 618:

4.4. Machine learning for spatially dense metrology

For spatially dense metrology across the entire wafer area, we employed a machine learning algorithm. The ridge linear regression, a regularization method that reduces overfitting by adding a penalty term to the least-squares cost function, was adopted. To train the regression model, reference measurement data, such as SEM measurements, were obtained, and 3×3 MM spectra were acquired using the IMMSE by averaging 15×15 pixels around the cell blocks nearest to each reference point. The entire dataset was divided into three groups—train, valid, and test set—for optimizing the regression model's performance with an appropriate regularization constant.

Line 625:

4.5. Machine learning for spatially dense metrology

For the spatially dense metrology across the entire wafer area, we employed a machine learning algorithm. Ridge regression, a regularized variant of least-squares regression, mitigates overfitting by incorporating a penalty term proportional to the squared magnitude of the coefficients. For model training, reference metrology data—such as SEM measurements—were collected from specific locations on the wafer. At each reference location, corresponding 3×3 MM spectra were obtained using the IMMSE system by averaging over 15×15 pixel regions centered on the nearest cell block.

The dataset consisting of paired MM spectra and reference values was partitioned into three subsets: training, validation, and test sets. This partitioning enabled proper optimization of the regression model, including the selection of an appropriate regularization constant. Once trained, the model was applied to the MM spectra acquired across the entire wafer, enabling spatially dense prediction of metrology data with more than 10 million measurement points across the wafer.

4) We revised the Discussion section to more effectively emphasize the novelty and impact of our work. The updated section begins by highlighting the improved spectrum consistency across the ultra-wide FOV, which serves as the foundation for achieving both high measurement accuracy and throughput. Building on this, we demonstrate that the IMMSE system successfully measured 12-inch DRAM wafers from a real high-volume production line, where the device structures and their process variations are extremely small—making accurate measurement highly challenging. The IMMSE achieved sub-nanometer accuracy in quantifying thin film thickness, CD, and overlay, while capturing process-induced variations across the wafer that are difficult to detect using conventional tools such as SEM.

We also addressed practical limitations in throughput and proposed several optimization strategies, such as selective wavelength sampling and reduced polarization conditions, to adapt the system for high-throughput process monitoring. Lastly, we briefly discussed the broader applicability of IMMSE beyond semiconductor metrology, including potential use in biomedical imaging, diagnostics, and analytical chemistry.

Line 387:

In this study, we experimentally demonstrated the ultra-wide-field imaging Mueller matrix spectroscopic ellipsometry (IMMSE) system for the semiconductor metrology. To achieve the wide-field metrology, we integrated a custom-designed optical system, which has a 20 mm × 20 mm field of view (FOV) with a spatial resolution of 6.5 μm. Using the IMMSE system, we measured hyperspectral image cubes, yielding over 10 million intensity spectra. For the Mueller matrix (MM) analysis, the hyperspectral image cubes were measured under multiple polarization conditions employing dual rotating polarizer configuration. By analyzing signal variations corresponding to changes in polarization states, 3×3 MM spectra from all pixels in the hyperspectral image cube were obtained.

To ensure reliable and consistent data acquisition across the wide FOV, we developed a unique algorithm to remove system-induced signal distortion. The algorithm, based on relative transmittance—referred to as system factors—compensates for signal distortions caused by spatial variations in the transmittance of optical components, such as rotating polarizers. For the clarity of the calculation, we utilized a matrix reformulation, which transforms matrix calculations of the intensity equation into scalar products.

For the experimental demonstration of the system-induced signal distortion removal, we measured the 3×3 MM spectrum of bare silicon wafer sample. After the system factors were removed, the root-mean-squared-error (RMSE) values relative to the theoretical spectrum were reduced to 2.6% of their values before removal. Similarly, the spectrum variations within the FOV were reduced to 4.7% of their values before removal.

The MM spectrum consistency across the FOV enabled spatially dense metrology for semiconductor manufacturing, providing numerous measurement points. To achieve the dense metrology, we employed a machine learning algorithm, specifically Ridge linear regression. To train the regression model, we utilized reference metrology data obtained from the conventional point-based method, such as scanning electron microscopy (SEM) and numerous MM spectra obtained using the IMMSE. The trained regression model yielded more than 10 million overlay values across the wafer, representing 12,800 times more data points compared to SEM. To evaluate the throughput of the IMMSE, we analyzed the point-by-point measurement time. The results showed that the IMMSE achieves 0.001 seconds per point, while the conventional SEM requires 0.7 seconds per point.

Leveraging its capability for the spatially dense metrology across the whole wafer area, we demonstrated that the IMMSE can be applied for the yield enhancement in semiconductor manufacturing. This was achieved by identifying spatial variations of dynamic random access memory (DRAM) structure within individual chips as well as across the wafer. The measurement results obtained using the IMMSE successfully identified distinctive spatial variation patterns in thin film thickness across the wafer. We presumed that the observed thickness variation patterns could be interpreted as fingerprints from the manufacturing equipment. We also demonstrated that these patterns disappeared under the revised process conditions. Since the conventional point-based ellipsometry method failed to detect these abnormalities, we confidently propose that the IMMSE could serve as a unique solution for optimizing the DRAM manufacturing processes.

Furthermore, we expect that the IMMSE can be utilized to evaluate photolithography processes. To verify feasibility, we measured the overlay misalignment error of a DRAM wafer. The overlay measurement results from the IMMSE clearly identified repetitive overlay variation patterns that were undetectable in the SEM results. The results offered significant insights into the early detection of defective chips, which would be

identified during the final electrical testing. Based on a numerous overlay values, we demonstrated that the IMMSE enables statistical analysis at the individual chip level, while SEM only provides fewer than two representative measurement values per chip. We showed that potential defective chips exceeding specific overlay limits can be rapidly identified from the statistical analysis.

Line 431:

In this study, we demonstrated an ultra-wide-field imaging Mueller matrix spectroscopic ellipsometry (IMMSE) system for semiconductor metrology, capable of acquiring over 10 million spatially resolved 3×3 Mueller matrix (MM) spectra across a $20 \text{ mm} \times 20 \text{ mm}$ field of view (FOV) with $6.5 \text{ }\mu\text{m}$ spatial resolution.

For the wide-FOV metrology, ensuring spectral consistency across the entire FOV area is critical. We achieved high spectrum consistency across the ultra-wide FOV, suppressing the spectrum error relative to the theoretical spectrum 94% and spectral variation across the FOV 73%. To enable this, we utilized a unique correction algorithm based on relative transmittance—referred to as system factors—which compensates for spatial non-uniformities of the signal induced from optical aberration and rotating polarizers.

Based on the high spectrum consistency across the FOV and the incorporation of a machine learning algorithm, the IMMSE enabled spatially dense metrology, revealing over 10 million metrology data across the wafer. Building upon these capabilities, practical applications of the IMMSE in DRAM manufacturing are suggested.

The experimental demonstrations were conducted on 12-inch DRAM wafers sourced from a real-world semiconductor manufacturing line, where the critical features are notably small and process variations are difficult to detect. The results highlight the IMMSE system's practical applicability and high sensitivity.

The thin film thickness, critical dimensions (CD), and overlay misalignment errors of the DRAM wafer were measured. The IMMSE enabled the detection of distinctive spatial variation patterns across the wafer, which were effectively suppressed under revised process conditions. These results demonstrate the effectiveness of the IMMSE in accurately monitoring and controlling process-induced pattern variations.

Additionally, statistical analysis of the overlay data enabled the identification of potentially defective chips exceeding the overlay thresholds. By employing the proposed technique, process abnormalities can be detected quickly and well before the electrical test, enabling rapid process feedback and contributing to yield improvement. These results suggest that the IMMSE system can serve as a powerful platform for evaluating and optimizing semiconductor manufacturing processes, ultimately contributing to yield enhancement.

The regression model employed in this study demonstrated high prediction accuracy, with root mean square error (RMSE) values below 1 nm for all evaluated parameters—including thin film thickness, CD, and overlay—as summarized in Table 3. This sub-nanometer accuracy indicates that the IMMSE system is capable of resolving nanometer-scale structural variations, which is essential for monitoring and controlling process-induced fluctuations in advanced semiconductor fabrication.

Table 3. DRAM metrology Accuracy of the IMMSE

	Film thickness		CD		Overlay	
	Process 1	Process 2	Process 1	Process 2	Process 1	Process 2
Accuracy (nm) (Reg. Model RMSE)	0.62	0.49	0.11	0.10	0.41	0.5

While the required accuracy may vary across applications, the achieved sub-nanometer precision is consistent with the metrology demands of next-generation semiconductor technologies, which continue to scale toward smaller nodes with tighter process tolerances⁵¹.

For the overlay metrology in this study, more than 10 million overlay values were generated across a single wafer—representing a 1,987-fold increase in data points compared to scanning electron microscopy (SEM). Each point was measured in 0.001 seconds, which is significantly faster than SEM and considered acceptable for practical use in semiconductor manufacturing (Table 4).

While IMMSE provides high-density metrology data with high throughput, improving throughput is essential in practical process monitoring. To address the practical need for faster wafer-level inspection in process

monitoring applications, several strategies can be considered to optimize the throughput of the IMMSE system. First, selective wavelength sampling can significantly reduce measurement time, focusing on a subset of wavelengths that are known to exhibit strong sensitivity to specific film parameters or structural variations. Second, the number of polarization conditions used for measurement can be adjusted based on the required sensitivity and complexity of the target parameter. For example, thin film thickness measurements often yield sufficient accuracy with four polarization states, and in some cases, monitoring trends over time may require only one or two states. However, it should be noted that certain applications such as overlay require 3×3 Mueller matrix reconstruction, and thus cannot be performed with reduced polarization sets. Finally, the number of measured shots across the wafer can be reduced by selecting representative regions, similar to the strategy used in conventional point-based metrology tools. This is especially effective in production environments where wafer uniformity is relatively high and full-wafer coverage is not always necessary. Together, these optimization strategies enable the IMMSE system to be tailored for high-throughput process monitoring while preserving its high-resolution and full-field measurement capabilities when needed.

Table 4. Throughput of conventional SEM and IMMSE. The throughput is calculated based on the measurement points over full wafer area and total measurement time.

	Measurement Points (pts)	Measurement Time (s)	Throughput (s/pts)
SEM	5,994	4,196	0.700
IMMSE	11,912,908	12,600	0.001

While the IMMSE system was developed primarily for semiconductor metrology, its underlying optical and analytical principles make it applicable to a range of scientific and clinical domains that benefit from wide-field, high-throughput polarimetric measurements. In particular, the ability to acquire spatially dense MM data over a broad spectral range and large FOV opens possibilities in life sciences, medical diagnostics, and analytical chemistry.

In its current configuration, the IMMSE system employs linear polarization states for both generation and analysis, yielding a reduced 3×3 subset of the full MM. Despite this limitation, prior studies have shown that essential polarization parameters—such as linear diattenuation, linear retardance, and linear depolarization—can be reliably extracted from this subset^{52,53}. These parameters have been applied in biomedical contexts, including the development of flexible fiber-based 3×3 MM probes for ex vivo tissue characterization⁵⁴, and in identifying optical signatures in retinal nerve fiber layers⁵⁵, visualizing anatomical features in brain tissue⁵⁶, and highlighting characteristic patterns in cancerous tissues⁵⁷.

These applications underscore the practical utility of partial MM measurements, especially in settings requiring compact instrumentation and simplified acquisition. Looking ahead, the IMMSE system’s modular design allows for potential expansion beyond linear polarization. For instance, integrating broadband wide-field waveplates could enable full 4×4 MM acquisition, granting access to circular dichroism and comprehensive polarization characterization. Realizing this capability would require precise calibration of the wavelength- and position-dependent retardance of the waveplates, but the in-situ polarization correction framework presented in this work offers a promising foundation for addressing these challenges.

51 IEEE International Roadmap for Devices and Systems (IRDS). More Moore. IEEE Int. Roadmap for Devices and Systems, (2018). <https://irds.ieee.org>

52 Ghosh, N., Wood, M. F. G., & Vitkin, I. A. Polar decomposition of 3×3 Mueller matrix: a tool for quantitative tissue polarimetry. *Opt. Express*, 14, 9324–9337 (2006).

53 He, H., Wang, Y., & Wang, Y. Study on the validity of 3×3 Mueller matrix decomposition. *J. Biomed. Opt.*, 20(6), 065003 (2015).

54 Alali, S., Vitkin, I. A., & Bizheva, K. Flexible polarimetric probe for 3×3 Mueller matrix measurements of biological tissue. *Sci. Rep.*, 7, 12099 (2017).

55 Bueno, J. M., & Campbell, M. C. W. Exploration of the retinal nerve fiber layer thickness by measurement of

the linear dichroism. *Appl. Opt.*, 44, 7074–7081 (2005).

56 Menzel, M., Reischl, L., Axer, M., & Amunts, K. Diattenuation of brain tissue and its impact on 3D polarized light imaging. *Biomed. Opt. Express*, 8, 3163–3178 (2017).

57 Ding, H., Wang, W., Wu, Y., Boppart, S. A., & Tang, Y. Mueller matrix polarimetry for differentiating characteristic features of cancerous tissues. *J. Biomed. Opt.*, 19(7), 076013 (2014).

5) To further clarify how wide-FOV metrology was achieved, we revised and expanded the relevant technical descriptions in Section 4.1 (Optical setup) and Section 2.3 (System factor analysis). On the hardware side, we implemented several key strategies, including the adoption of an Offner-type reflective imaging system to minimize aberrations, a low-NA illumination design to ensure uniformity with minimal spectral degradation, and application of the Scheimpflug condition to maintain sharp focus across the ultra-wide FOV. We also introduced a precision-adjustable camera mount to finely align the sensor position and keep all regions within the depth of focus.

On the software side, we applied camera linearity correction to ensure consistent intensity response and implemented distortion correction algorithms to reduce geometric artifacts and preserve spatial accuracy. These combined efforts enhance signal fidelity and support reliable, high-accuracy metrology across the entire imaging area.

Line 515:

An Offner configuration, comprising a pair of concentric spherical mirrors, was employed as an imaging optics to minimize image distortion and chromatic aberration⁴²⁻⁴⁴ across the wide FOV of 20 mm × 20 mm and a broadband wavelength range of 250 nm to 1100 nm. The numerical aperture (NA) of the imaging optics was designed to be 0.06, considering the trade-off between the overall size of the optical system and its spectral sensitivity. A back-illuminated scientific complementary metal oxide semiconductor (sCMOS) image sensor was used to record signals reflected from the sample wafer. The position of the image sensor was precisely adjusted using a custom-designed 6-axis mechanical stage to satisfy the Scheimpflug imaging condition for the oblique incident angle of 65°.

Line 523:

The imaging module of the proposed IMMSE system was designed to achieve an ultra-wide FOV while maintaining high optical performance and minimizing chromatic aberration. The imaging optics adopted a multi-mirror reflective configuration, known as Offner configuration, which inherently minimizes chromatic dispersion and imaging distortion across a broad wavelength range. The Offner relay was configured with a numerical aperture (NA) of 0.06 and a telecentricity angle below 0.01°, balancing the trade-off between optical resolution and the system's sensitivity to nanoscale structural variations.

Although the imaging optics employed a reflective configuration, residual chromatic aberration was unavoidable due to the presence of the wire-grid polarizer and the image sensor's window glass. To minimize image distortion across the wide FOV, the curvature ratio between two concentric spherical mirrors of the Offner relay was precisely adjusted in design. By tuning the mirror curvatures, the image distortion was reduced to less than 0.19% across the entire FOV and the residual chromatic aberration was estimated as a focal plane shift of less than 40 μm across the system's full wavelength range.

Due to the oblique illumination geometry with a 65° incidence angle, the image plane is inherently tilted with respect to the sensor plane. Under the Scheimpflug condition, the required sensor tilt increases with magnification, which can significantly reduce light collection efficiency and complicate mechanical alignment. To avoid these issues, a 1× magnification was selected and minimized the necessary sensor tilt while preserving optical performance. The position and orientation of the image sensor were then precisely adjusted using a custom-designed 6-axis mechanical stage to satisfy the Scheimpflug condition. This ensured that the entire imaging plane was maintained within the system's depth of focus, thereby minimizing defocus-induced measurement errors.

Line 118:

2.2. System-induced signal distortion

One of the key features of the proposed wide-field IMMSE system for semiconductor manufacturing is its ability to deliver consistent and reliable measurement results over the FOV. Specifically, measurements taken at the same point on a sample should remain consistent, whether the point is located at the center or the edge of the FOV. However, in practice, signal distortion occurs depending on the positions within the FOV.

The measurement discrepancies within the large FOV are expected to arise from the non-uniformity of optical performance over their operational regions, such as the transmittance or reflectance of lenses, mirrors and polarizers. Specifically, the signal recorded at each pixel is the result of light propagated through different

regions of optical components. These components exhibit position- and wavelength-dependent variations in transmittance and reflectance, which leads to pixel-wise discrepancies in the measured intensity. In addition to the performance inconsistency of the optical components, the signal also changes with the rotation of the polarizer, as the light passes through different areas of polarizer by the rotation, as shown in Fig. 2a.

Previous research on the imaging ellipsometry addressed the inhomogeneity of retardation in rotating compensator by employing an extra phase constant in the equation^{34,35}. In this work, we introduce a novel method to obtain reliable signals across the FOV by eliminating system-induced signal distortion in the dual rotating polarizer configuration. This was achieved by employing condition-specific relative transmittance (RT) constant and compensating for the performance changes in the PSG, PSA, and wavelength.

In addition to polarization-induced intensity variations, residual geometric distortion from the imaging optics can degrade the spatial alignment of hyperspectral data across the wide FOV. Although the Offner-type reflective relay used in the imaging module minimizes chromatic and geometric aberrations, slight spatial distortion remains due to practical implementation factors. To address this, we applied a geometric distortion correction during post-processing. A reference target was used to measure the pixel-wise displacement vectors across the field, and these vectors were used to remap each image to a common spatial grid. In semiconductor metrology, even a few micrometers of spatial misalignment can affect spectral accuracy, especially when analyzing pixel-wise variations across polarization conditions. Our correction ensures spatial consistency across wavelengths and polarization states, supporting accurate Mueller matrix reconstruction at high spatial resolution.

Furthermore, while the camera manufacturer provides basic linearity correction, some residual nonlinear response remains, particularly in low-intensity regions. This is critical in our system due to large intensity variation across polarization states. Additional calibration was performed using a bare Si wafer under 500 nm illumination, with the exposure time incrementally increased by 0.015 ms. A separate dark measurement was also taken. The resulting intensity curve was corrected using spline interpolation to generate a refined response map.

Line 151:

The intensity and partial polarization characteristics of light are described using the Stokes vector. The relationship between the light before and after interacting with the sample can be expressed as follows:

$$\vec{s}_{\text{out}} = M_1 M_2 \cdots M_n \vec{s}_{\text{in}} \quad (1)$$

where \vec{s}_{in} and \vec{s}_{out} are Stokes vectors of input and output light, respectively, M_1, M_2, \dots, M_n are the MMs of optical components in the system. Since the dual rotating polarizer configuration was employed in this study, the MM is represented as a 3×3 square matrix, excluding circular polarization characteristics⁴⁵. Based on Eq. (1), the intensity measured using the proposed system can be expressed as a matrix product of MMs of the polarization manipulation components and the sample wafer, as shown below:

$$L_{ij} = \vec{d}^T Q_j M P_i \vec{s} \quad (2)$$

where L_{ij} is the measured intensity at the i -th PSG condition and the j -th PSA condition. M represents the MM of the sample wafer, P_i and Q_j denote the MMs of the PSG and the PSA, respectively. \vec{s} is Stokes vector of illumination light and \vec{d} represents the cumulative polarization sensitivity of all detection-side components after the PSA. This includes the Offner relay, sensor window, and the sCMOS image sensor, all modeled as Mueller matrices. To incorporate the RT of the optical system, we reformulated both polarizer MMs P_i and Q_j as follows:

Reviewer #4 (Remarks to the Author):

The authors should add the following information to the manuscript:

1. The brand of the polarizers used, or at least their extinction ratio.

We appreciate the reviewer's suggestion to specify the brand and extinction ratio of the polarizer. In our system, polarization control and analysis are achieved using a high-performance wire-grid polarizer manufactured by Moxtek (US). This component offers an extinction ratio greater than 800 even at wavelengths below 300 nm, while maintaining a transmittance of approximately 62%, ensuring both polarization purity and efficient light throughput. This information has been added to the section 4.1 Optical setup in the revised manuscript.

Line 555:

The wire-grid polarizer is one of the most critical optical components in the IMMSE system. It was used to manipulate the polarization state of the illumination light and analyze the polarization sensitivity of the measured structures. To achieve this, we employed a high-performance wire-grid polarizer (Moxtek, US), which exhibits an exceptionally high extinction ratio. Notably, the polarizer maintains an extinction ratio greater than 800 even at wavelengths below 300 nm, ensuring reliable polarization contrast in the short-wavelength region. In addition, it offers a transmittance level of approximately 62%, enabling efficient light throughput. These characteristics are essential for preserving the accuracy of polarization-resolved measurements across the FOV and a broad spectral range.

2. The resolution of the analog-to-digital (A/D) conversion of the images acquired by the sCMOS image sensor used in the IMMSE system.

We appreciate the reviewer's comment and have revised the manuscript to explicitly state that the scientific CMOS (sCMOS) image sensor integrated into the IMMSE system supports 16-bit analog-to-digital (A/D) conversion, corresponding to 65,536 discrete intensity levels per pixel.

In addition to clarifying the A/D conversion resolution, we have also expanded the description to include other key performance specifications of the sensor. These include the resolution (3200 × 3200 pixels), pixel pitch (6.5 μm), active sensing area (20.8 mm × 20.8 mm), full well capacity (15,000 e⁻), read noise (1.6 e⁻), conversion gain (0.23 e⁻/count), and dark current (1.27 e⁻/pixel/sec at 0 °C) under active air cooling. Furthermore, we have specified the quantum efficiency performance, which exceeds 30% in the short-wavelength region and reaches up to 95% in the visible spectrum. These comprehensive specifications highlight the sensor's capability for high dynamic range, low-noise performance, and high sensitivity across the entire FOV.

Line 542:

The detector integrated into the system is a scientific CMOS (sCMOS) sensor featuring a resolution of 3200 × 3200 pixels with a pixel pitch of 6.5 μm, yielding an active sensing area of 20.8 mm × 20.8 mm. The sensor supports 16-bit analog-to-digital conversion, enabling 65,536 discrete intensity levels per pixel. The quantum efficiency of the sensor exceeds 30% in the short-wavelength region and reaches up to 95% within the visible spectrum, ensuring high photon-to-electron conversion efficiency across a broad spectral range. With a full well capacity of 15,000 electrons, conversion gain of 0.23 e⁻/count, and a low read noise of 1.6 e⁻, the system ensures high dynamic range and fine quantization accuracy. Active air cooling maintains the dark current at 1.27 e⁻/pixel/sec at 0 °C, supporting low-noise, high-speed imaging performance.

In my opinion, this work is undoubtedly of high scientific quality and quite innovative. However, I do not believe it has broad enough scientific relevance to justify publication in Nature Communications.

There are numerous applications of polarimetry in the life sciences and chemistry (e.g., detection of food adulteration, chemical and biological analysis, etc.), and others have been suggested (e.g., identification of metastatic tissues and viruses). However, in all these cases, it is essential to measure quantities such as the degree of polarization and dichroism, capabilities that, in my view, the IMMSE system in its current optical configuration cannot provide.

We sincerely thank the reviewer for their thoughtful and constructive feedback. We fully acknowledge the importance of capturing comprehensive polarization information — including the full degree of

polarization and circular dichroism — particularly for advanced applications in life sciences and chemistry involving chiral molecules and optically active media.

While the current optical configuration of the IMMSE system, based on two rotating polarizers, does not enable full 4×4 Mueller matrix (MM) acquisition or direct measurement of circular dichroism, we would like to respectfully clarify that the 3×3 MM subset still provides meaningful and practically useful information. Through established polar decomposition techniques [1,2], it is possible to estimate linear diattenuation, linear depolarization, linear retardance, and limited circular retardance. Although these estimates are derived from partial data and should be interpreted with caution, they have been successfully applied in various biomedical and analytical contexts.

For example, 3×3 MM measurements have been directly implemented in fiber-based tissue probes [3]. Additionally, polarization parameters such as linear diattenuation, linear retardance, and linear depolarization — which can be estimated from 3×3 MM — have been utilized in diverse biomedical applications, including retinal nerve fiber analysis for glaucoma detection [4], anatomical mapping of brain structures [5], and differentiating characteristic features of cancerous tissues [6].

These examples demonstrate that even partial MM analysis can provide significant value in applied biomedical contexts, particularly when high-throughput measurement is desired. In this regard, we believe the IMMSE system, despite its current focus on semiconductor metrology, offers potential relevance to broader scientific domains that utilize linear polarization-based contrast mechanisms.

That said, we fully acknowledge the reviewer's point: the current IMMSE configuration does not allow for full MM acquisition, and thus cannot provide complete access to circular dichroism or the full degree of polarization — quantities that may be essential in specific applications within life sciences and chemistry.

While the present study does not address these parameters, the IMMSE platform is optically compatible with future extensions to support full 4×4 MM measurements. The current Offner relay-based design enables large-area and broadband imaging, and could be modified to incorporate broadband wide-field waveplates. Doing so would require accurate calibration of their wavelength- and position-dependent retardance, which is technically challenging. However, we believe that the in-situ polarization correction approach introduced in this work can be extended to manage these additional calibration complexities.

We include this as a prospective development direction and appreciate the reviewer's comments, which helped us refine our understanding of the broader application space and long-term potential of the IMMSE system.

These details are now explicitly described in the revised Discussion section of the manuscript.

Line 481:

While the IMMSE system was developed primarily for semiconductor metrology, its underlying optical and analytical principles make it applicable to a range of scientific and clinical domains that benefit from wide-field, high-throughput polarimetric measurements. In particular, the ability to acquire spatially dense MM data over a broad spectral range and large FOV opens possibilities in life sciences, medical diagnostics, and analytical chemistry.

In its current configuration, the IMMSE system employs linear polarization states for both generation and analysis, yielding a reduced 3×3 subset of the full MM. Despite this limitation, prior studies have shown that essential polarization parameters—such as linear diattenuation, linear retardance, and linear depolarization—can be reliably extracted from this subset [1,2]. These parameters have been applied in biomedical contexts, including the development of flexible fiber-based 3×3 MM probes for ex vivo tissue characterization [3], and in identifying optical signatures in retinal nerve fiber layers [4], visualizing anatomical features in brain tissue [5], and highlighting characteristic patterns in cancerous tissues [6].

These applications underscore the practical utility of partial MM measurements, especially in settings requiring compact instrumentation and simplified acquisition. Looking ahead, the IMMSE system's modular design allows for potential expansion beyond linear polarization. For instance, integrating broadband wide-field waveplates could enable full 4×4 MM acquisition, granting access to circular dichroism and comprehensive

polarization characterization. Realizing this capability would require precise calibration of the wavelength- and position-dependent retardance of the waveplates, but the in-situ polarization correction framework presented in this work offers a promising foundation for addressing these challenges.

Theory:

[1] Polar decomposition of 3x3 Mueller matrix: a tool for quantitative tissue polarimetry (2006), Optics Express, <http://dx.doi.org/10.1364/OE.14.009324>

[2] Study on the validity of 3 × 3 Mueller matrix decomposition (2015), Journal of Biomedical Optics, <https://doi.org/10.1117/1.JBO.20.6.065003>

Experiment:

[3] Flexible polarimetric probe for 3x3 Mueller matrix measurements of biological tissue (2017), Scientific Reports, <https://doi.org/10.1038/s41598-017-12099-8>

[4] Exploration of the retinal nerve fiber layer thickness by measurement of the linear dichroism (2005), Applied Optics, <https://doi.org/10.1364/AO.44.007074>

[5] Diattenuation of brain tissue and its impact on 3D polarized light imaging (2017), Biomedical Optics Express, <https://doi.org/10.1364/BOE.8.003163>

[6] Mueller matrix polarimetry for differentiating characteristic features of cancerous tissues (2014), Journal of Biomedical Optics, <https://doi.org/10.1117/1.JBO.19.7.076013>

Reviewer #1 (Remarks to the Author):

I think the authors made a good revision and addressed all my concerns on the manuscript. I think it can be accepted now. I just have only one optional suggestion that whether the authors can provide the repeatability at the same location by the proposed instrument and method, which is another critical index in volume IC manufacturing.

We sincerely appreciate the reviewer's thoughtful comments and positive evaluation of our revised manuscript. We also thank you for the additional suggestion regarding repeatability at the same measurement location, which is indeed an important performance metric in high-volume IC manufacturing environments.

In response, we have now included repeatability measurements using the proposed IMMSE system, demonstrating its measurement consistency. Specifically, we added a new paragraph in the Discussion section summarizing the results of repeated measurements at the same location across multiple acquisitions. This analysis confirms that the IMMSE provides highly consistent measurements, achieving a 3-sigma repeatability of 0.36 nm, and supports its applicability in semiconductor manufacturing. Additionally, the Methods section has been expanded to describe the experimental procedures for acquiring the repeatability data, including the number of repetitions.

We hope this addition fully addresses the reviewer's suggestion, and once again thank you for your valuable feedback.

Line 343:

As a further evaluation of system performance, we assessed repeatability of the system by conducting repeated overlay measurements at the same wafer location. The resulting 3-sigma repeatability, averaged across all measurement points within a single shot, was approximately 0.36 nm. This indicates stable and consistent overlay measurement capability, making the IMMSE system suitable for high-volume semiconductor manufacturing (See Methods for detailed evaluation conditions).

Line 550:

4.6 Overlay measurement repeatability

To verify the repeatability of the IMMSE system, the overlay between the bit line (BL) and active (ACT) patterns on a DRAM sample wafer was measured. A total of seven repeated measurements were conducted at the same wafer over a period of approximately 200 hours, under identical system and environmental conditions. The system-induced signal distortion of each repeated measurement was corrected using the system factors defined before the initial measurement, to verify that the system factors remain robust under environmental fluctuations.

The regression model was trained using identical reference overlay values and the IMMSE spectra obtained individually from each repeated measurement. For the repeatability analysis, the standard deviation of repeated measurements was first calculated at each of the 12,672 measurement points within a single exposure shot located at the wafer center. The 3-sigma repeatability was then obtained by averaging these standard deviations across all measurement points in the shot area. The resulting repeatability was 0.36 nm, demonstrating that the IMMSE system maintains highly stable performance for sub-nanometer process control and monitoring in semiconductor manufacturing over extended periods.

Reviewer #2 (Remarks to the Author):

The authors have addressed the comments and questions from this reviewer in a satisfactory way, therefore the revised manuscript is recommended to be accepted for publication.

Reviewer #3 (Remarks to the Author):

I thank the authors for their efforts to respond to my concerns. The comparison of representative imaging Mueller matrix ellipsometry systems (Table 3. in the response to my comments) and relative comments on the corresponding references (also one more ref. is recommended Opt. Express 29: 32712-32727, 2021) should be added in the manuscript to clearly talk about the state-of-art of IMMSE and the contribution of this work to the community.

We sincerely thank the reviewer for the constructive feedback and kind support. Following the suggestion, we have added to the Discussion section a comparison table of representative imaging Mueller matrix ellipsometry systems, along with the corresponding comments. The reviewer-recommended reference [Opt. Express 29, 32712–32727 (2021)] has also been included in this table. In addition, the descriptions of the “Inhomogeneity correction items” for the representative systems have been updated to allow for comprehensive comparison.

This addition enables a clear positioning of our work within the current state-of-the-art, in line with your thoughtful suggestion.

Line 369:

Previous studies have made significant advances in addressing specific optical imperfections in ellipsometry systems. These include the phase retardation of rotating compensators⁵², the initial angular offsets of polarizers and compensators across wavelengths⁵³, nulling-based optimization approaches⁵⁴, and selective system calibration procedures for ellipsometry system, including polarizers and compensators⁵⁵.

These works have primarily been demonstrated on standardized samples or through contrast-based imaging, and their calibration strategies have generally targeted specific components or error sources⁵²⁻⁵⁶. In this context, the IMMSE employs a generalized calibration framework that corrects signal distortions from all optical components in the illumination path, imaging optics, and polarizers by using the relative transmittance. This approach comprehensively accounts for the entire optical system, ensuring consistent and reliable measurements across the wide field of view and broad spectral range. This capability is particularly advantageous for accurately characterizing sub-nanometer structural variations in DRAM wafers.

Line 802:

52. Jin, L., Iizuka, Y., Iwao, T., Kondoh, E., Uehara, M. & Gelloz, B. Calibration of the retardation inhomogeneity for the compensator-rotating imaging ellipsometer. *Appl. Opt.* **58**, 9224–9229 (2019).

53. Chen, X. et al. Development of a spectroscopic Mueller matrix imaging ellipsometer for nanostructure metrology. *Rev. Sci. Instrum.* **87**, 053113 (2016).

54. Braeuninger-Weimer, P. et al. Fast, noncontact, wafer-scale, atomic layer resolved imaging of two-dimensional materials by ellipsometric contrast micrography. *ACS Nano* **12**, 8555–8563 (2018).

55. Chen, C., Chen, X., Wang, C., Sheng, S., Song, L., Gu, H. & Liu, S. Imaging Mueller matrix ellipsometry with sub-micron resolution based on back focal plane scanning. *Opt. Express* **29**, 32712–32727 (2021).

56. Sigger, F. et al. Spectroscopic imaging ellipsometry of two-dimensional TMDC heterostructures. *Appl. Phys. Lett.* **121**, 071101 (2022).

Table 3. Comparison of representative imaging Mueller matrix ellipsometry systems. This table compares key features of the proposed system and three representative imaging Mueller matrix ellipsometry systems reported in the literature.

	This work	Jin et al. [1]	Chen et al. [2]	Braeuninger-Weimer et al. [3]
System Configuration	Dual rotating polarizer (3×3 MM)	PCrSA (Δ, ψ)	Dual rotating compensator (4×4 MM)	Fixed PCSA (Film model parameters)
Demonstrated Wavelength (nm)	250-1100	650	400-700	390, 450, 550
Field of View (mm ²)	20×20	5.7×4.3	4×4	2×2
Residual Spectral Error	0.004	-	0.01	-
Inhomogeneity Correction Items	Polarizer transmittance Polarization sensitivity of detection path Polarization state of illumination on PSA	Retardation	- (Pixel averaged system parameters)	-

Table 4. Comparison of representative imaging Mueller matrix ellipsometry systems. Key features of the proposed system and four representative imaging Mueller matrix ellipsometry systems reported in the literature are compared in the table.

	This work	Jin et al. [52]	Chen et al. [53]	Braeuninger-Weimer et al. [54]	Chao et al. [55]
System Configuration	Dual rotating polarizer (3×3 MM)	PCrSA (Δ, ψ)	Dual rotating compensator (4×4 MM)	Fixed PCSA (Film model parameters)	Dual rotating compensator (4×4 MM)
Demonstrated Wavelength (nm)	250-1100	650	400-700	390, 450, 550	400-700
Field of View (mm ²)	20×20	5.7×4.3	4×4	2×2	-
Residual Spectral Error	0.004	-	0.01	-	-
Inhomogeneity Correction Items	Comprehensive relative transmittance	Compensator	Polarizer and compensator	Polarizer and compensator	Polarizer, compensator, beam splitter, beam splitter, and objective lens

Due to the journal's policy limiting the number of items, the original Table 1 and Table 2 have been merged into a single Table 1, and the newly added comparison table appears as Table 4.

Table 1. Spectrum errors with respect to the theoretical spectrum. Spectrum error of (A) in Fig. 3b. (I) and (II) indicate the spectrum error values for both before and after the system factors removal, respectively.

	m12	m13	m21	m22	m23	m31	m32	m33	
A	I	0.1126	0.0204	0.2283	0.2089	0.0160	0.0348	0.0088	0.1214
	II	0.0085	0.0009	0.0086	0.0064	0.0013	0.0009	0.0008	0.0076

Table 2. Variation of spectrum errors within FOV. The spectrum error variations within the FOV (B) before and (C) after the system factors removal, respectively.

	m12	m13	m21	m22	m23	m31	m32	m33
B	0.0018	0.0040	0.0025	0.0039	0.0036	0.0073	0.0014	0.0029
C	0.0010	0.0003	0.0006	0.0020	0.0002	0.0003	0.0002	0.0015

Table 1. Spectrum errors with respect to the theoretical spectrum and within FOV. (A) in Fig. 3b and the table below shows the spectrum error relative to the theoretical spectrum. (I) and (II) indicate the spectrum error values for both before and after the system factors removal, respectively. The spectrum error variations within the FOV (B) before and (C) after the system factors removal, respectively.

	m12	m13	m21	m22	m23	m31	m32	m33	
A	I	0.1126	0.0204	0.2283	0.2089	0.0160	0.0348	0.0088	0.1214
	II	0.0085	0.0009	0.0086	0.0064	0.0013	0.0009	0.0008	0.0076
B	0.0018	0.0040	0.0025	0.0039	0.0036	0.0073	0.0014	0.0029	
C	0.0010	0.0003	0.0006	0.0020	0.0002	0.0003	0.0002	0.0015	

Line 208:

Table 1 presents the summary of spectrum errors both before and after the system factor removal. The result verified that the spectrum error values were reduced to less than 10^{-2} for all MM elements after the system factors were removed.

Line 210:

(A) in Table 1 presents the summary of spectrum errors relative to the theoretical spectrum, both before and after the system factor removal. The result verified that the spectrum error values were reduced to less than 10^{-2} for all MM elements after the system factors were removed.

Line 217:

For the quantitative evaluation, the standard deviation of the spectrum error maps of all MM components for both (B) and (C) were calculated as shown in Table 12.

The following changes are unavoidable renumbering of existing tables due to the new table.

Table 3. DRAM metrology Accuracy of the IMMSE

	Film thickness		CD		Overlay	
	Process 1	Process 2	Process 1	Process 2	Process 1	Process 2
Accuracy (nm) (Reg. Model RMSE)	0.62	0.49	0.11	0.10	0.41	0.5

Table 2. DRAM metrology Accuracy of the IMMSE

	Film thickness		CD		Overlay	
	Process 1	Process 2	Process 1	Process 2	Process 1	Process 2
Accuracy (nm) (Reg. Model RMSE)	0.62	0.49	0.11	0.10	0.41	0.5

Line 335:

The regression model employed in this study demonstrated high prediction accuracy, with root mean square error (RMSE) values below 1 nm for all evaluated parameters—including thin film thickness, CD, and overlay—as summarized in Table 32.

Table 4. Throughput of conventional SEM and IMMSE. The throughput is calculated based on the measurement points over full wafer area and total measurement time.

	Measurement Points (pts)	Measurement Time (s)	Throughput (s/pts)
SEM	5,994	4,196	0.700
IMMSE	11,912,908	12,600	0.001

Table 3. Throughput of conventional SEM and IMMSE. The throughput is calculated based on the measurement points over full wafer area and total measurement time.

	Measurement Points (pts)	Measurement Time (s)	Throughput (s/pts)
SEM	5,994	4,196	0.700
IMMSE	11,912,908	12,600	0.001

Line 349:

Each point was measured in 0.001 seconds, which is significantly faster than SEM and considered acceptable for practical use in semiconductor manufacturing (Table 43).

Reviewer #4 (Remarks to the Author):

I sincerely thank the authors for significantly improving the manuscript and answering my questions in a more than satisfactory manner. In its current form, I believe that the manuscript is certainly suitable for publication in Nature Communications.

Minor corrections

2.1

Line 80:

Fig. 1a shows the schematic of the proposed IMMSE system, comprising three main modules: illumination module, imaging module, and polarization manipulation module.

Line 82:

A plasma-based broadband light source was employed to cover the system's wide wavelength range, extending from ultraviolet to infrared range.

Line 97:

For the experimental demonstration, we developed a metrology equipment that was integrated with the optical setup, a wafer transport robot, and a stage system for wafer movement.

Line 105:

The obtained signals were calibrated using unique data processing algorithm to correct for signal distortion induced by the measurement system, including rotating polarizer and aberrations, including such as image distortion.

Line 109:

This approach enabled high-spatial-density, wafer-scale metrology of nanostructures.

2.2

Line 117:

such as the transmittance or reflectance of lenses, mirrors, and polarizers.

Line 135:

Our correction ensures spatial consistency across wavelengths and polarization states, supporting accurate Mueller matrix MM reconstruction at high spatial resolution.

Line 140:

Additional calibration was performed using a bare Si-silicon wafer under 500 nm illumination

2.3

Line 159:

are the i -th rotation angle of the PSG and the j -th rotation angle of the PSA, respectively.

Line 155:

This includes the Offner relay, sensor window, and the sCMOS image sensor, all modeled as ~~Mueller matrices~~ MMs.

Line 164:

represents PSG-side total RT, defined as the PSG factor and ρ .

Line 177:

Fig. 2c shows the calculated PSG factor at a wavelength of 405 nm under four of the 36 rotation conditions, specifically when the PSG was rotated to 0°, 90°, 180°, and 270°.

2.4

Line 205:

Any nonzero values indicate the residual of the system-induced signal distortions. For the off-diagonal components, such as m_{13} , m_{31} , m_{23} , and m_{32} , the MM values exhibited approximately zero after the system factors were removed.

2.5

Line 223:

To enable spatially dense metrology based on these spectra, we employed a machine learning algorithm—specifically, Ridge regression—~~which was trained using both the acquired massive number of MM spectra and reference data obtained from conventional point based metrology tools such as SEM.~~ The regression model was trained using both the acquired massive number of MM spectra and reference data obtained from conventional point-based metrology tools such as SEM.

Line 234:

As illustrated in Fig. 4a, the schematic of the CVD setup shows thin-film deposition inside a gas chamber equipped with multiple nozzles arranged in a hexagonal array, which affects the spatial uniformity during deposition.

Line 236:

As illustrated in Fig. 4a, the CVD schematic shows thin-film deposition within a gas chamber with multiple nozzles arranged in a hexagonal array, which can affect the spatial thickness uniformity during deposition.

Line 239:

~~Figure 4b~~ Fig. 4b presents the thickness measurement results of a wafer processed under a specific CVD condition

Line 249:

Unlike Fig. 4c, Fig. 4d exhibits no distinctive patterns, implying that the modified process conditions are better suited for this manufacturing step.

Line 277:

The combination of ultra-small feature sizes and sub-nanometer misalignment

Line 300:

Fig. 6k illustrates the distribution of the mean and standard deviation of overlay values computed from all cell blocks within each of 100 chips located in the regions shown in Fig. 6f and 6i.

Line 302:

The result indicates that the distribution of both ~~metrics in Process 2 is significantly improved compared to those in Process 1. mean and standard deviation, in Process 2 are significantly improved compared to those in Process 1.~~

Discussion

Line 309:

We achieved high spectrum consistency across the ultra-wide FOV, suppressing the spectrum error relative to the theoretical spectrum 94% and spectral variation across the FOV 73%.

Line 311:

We achieved high spectrum consistency across the ultra-wide FOV, suppressing the spectrum error relative to the theoretical MM spectrum of bare silicon wafer by 94% and spectral variation across the FOV by 73%, compared to the conventional MM calculation method.

Line 319:

Building upon these capabilities, practical applications of the IMMSE in DRAM manufacturing ~~are~~ were suggested.

Line 320:

The experimental demonstrations were conducted on 12-inch DRAM wafers sourced from a real-world semiconductor manufacturing line fab

Line 323:

The thin film thickness, critical dimensions (CD), and overlay misalignment errors of the DRAM wafer were measured. The IMMSE enabled the detection of distinctive spatial variation patterns across the wafer, which were effectively suppressed under revised process conditions.

Line 325:

The thin film thickness, critical dimensions (CD), and overlay misalignment errors of the DRAM wafer were measured using the propose method. The IMMSE enabled the detection of distinctive spatial variation patterns across the wafer, which can be effectively suppressed under revised process conditions.

Line 333:

These results ~~suggest~~ confirm that the IMMSE system can serve as a powerful platform for evaluating and optimizing semiconductor manufacturing processes, ultimately contributing to yield enhancement.

Line 360:

However, it should be noted that certain applications such as overlay require 3×3 ~~Mueller matrix~~ MM reconstruction,

Line 364:

Together, these optimization strategies enable the IMMSE system to be tailored for high-throughput process monitoring while preserving its high-resolution and full-field measurement capabilities when needed.

Line 366:

Collectively, these optimization strategies allow the IMMSE system to be flexibly configured for high-throughput process monitoring, while retaining its capability for high-resolution, full-field measurements when required.

4.4

Line 495:

At each quadrant, a ~~Mueller matrix~~ MM spectroscopic image is captured.

Fig. 6 Caption

Line 896:

Distribution of the mean and standard deviation of overlay values for 100 chips located within the region indicated in (f) and (i), showing statistical improvement in overlay control under Process 2.

Line 898:

Statistical distribution of the mean and standard deviation of overlay values for 100 chips within the regions marked in (f) and (i), demonstrating improved overlay control under Process 2.